# Leveraging Hyperbolic Embeddings for Coarse-to-Fine Robot Design

**Heng Dong**[1*]  **Junyu Zhang**[2*]  **Chongjie Zhang**[3]

[1] IIIS, Tsinghua University   [2] Huazhong University of Science and Technology
[3] Washington University in St. Louis
{drdhxi,jyzhang1208}@gmail.com, chongjie@wustl.edu

## Abstract

Multi-cellular robot design aims to create robots comprised of numerous cells that can be efficiently controlled to perform diverse tasks. Previous research has demonstrated the ability to generate robots for various tasks, but these approaches often optimize robots directly in the vast design space, resulting in robots with complicated morphologies that are hard to control. In response, this paper presents a novel coarse-to-fine method for designing multi-cellular robots. Initially, this strategy seeks optimal coarse-grained robots and progressively refines them. To mitigate the challenge of determining the precise refinement juncture during the coarse-to-fine transition, we introduce the Hyperbolic Embeddings for Robot Design (HERD) framework. HERD unifies robots of various granularity within a shared hyperbolic space and leverages a refined Cross-Entropy Method for optimization. This framework enables our method to autonomously identify areas of exploration in hyperbolic space and concentrate on regions demonstrating promise. Finally, the extensive empirical studies on various challenging tasks sourced from EvoGym show our approach's superior efficiency and generalization capability.

## 1 Introduction

For decades, humans have envisioned the creation of artificial creatures with morphological intelligence (Lipson & Pollack, 2000; Howard et al., 2019; Gupta et al., 2021b). To achieve this goal, a highly promising solution is to learn optimal robot morphologies for a variety of tasks in simulated environments (Sims, 1994a; Wang et al., 2019; Yuan et al., 2021; Dong et al., 2023). Inspired by the wide existence of multi-cellular organisms, such as animals, land plants, and most fungi, one natural way of representing robots is to formulate them as multi-cellular systems. The creation of multi-cellular robots is challenging because of the twin difficulties: 1) the design space of robots, including cell types, positions, and parameters, *etc.*, is immensely large, and 2) the evaluation of each design requires learning its optimal control policy, which is often computationally expensive.

For multi-cellular robot design, previous methods typically adopt Genetic Algorithms (GA) (Sims, 1994b; Medvet et al., 2021; Cheney et al., 2014b), which sample robots from a large population to accomplish the given tasks and then only keep the top-performing robots and their offspring. However, these methods are inclined to learn from scratch in the vast design space and could be highly sample-inefficient (Wang et al., 2022). Intuitive examples are provided in Figure 1, where we use time-lapse images to show the control process of different designs performing task ObstacleTraverser-v0 (Bhatia et al., 2021). This task requires 2D robots to cross the terrain that is full of obstacles and is getting increasingly bumpy. GA (Figure 1 (a-b)) fails to learn any effective robot structures to solve this task because of the vast design space. To overcome this issue, Wang et al. (2022) uses a curriculum-based design (CuCo) method by gradually expanding the robot design from a small size to the target size through a predefined curriculum. However, as shown in Figure 1 (c,d) and empirical evidence in Section 5, robots of small size can hardly solve the original task and thus, unfortunately, cannot provide helpful information for the next stage of the curriculum.

In view of these challenges, we propose to design multi-cellular robots in a coarse-to-fine manner by first searching for coarse-grained robots with satisfactory performance and subsequently refining

---

*Equal contributions.

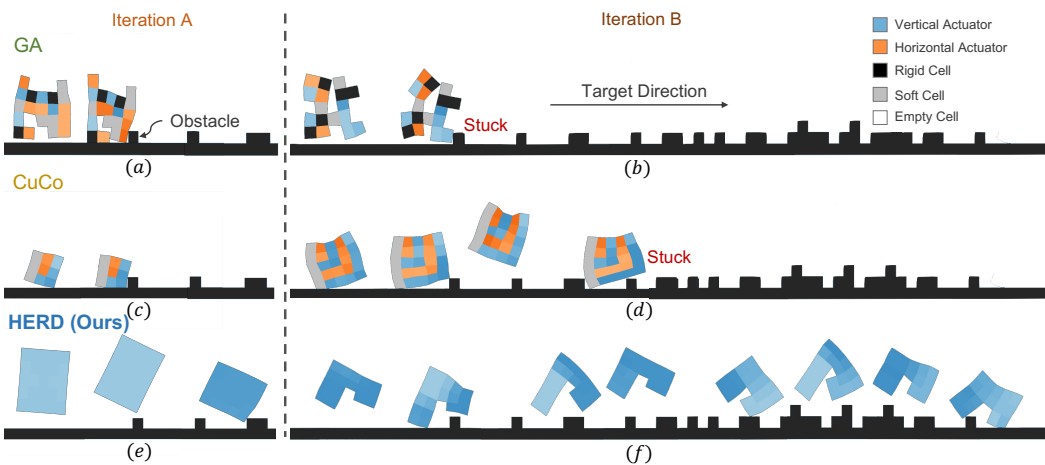

Figure 1: Visualization of the control process of our method HERD compared with baseline GA and CuCo (Wang et al., 2022) during training in task ObstacleTraverser-v0 (Bhatia et al., 2021). Iteration B is one of the iterations later than Iteration A, such that corresponding algorithms have updated the robot. (a-b) GA directly searches in the vast design space and fails to learn effective structures to cross the obstacles; (c-d) CuCo adopts a predefined curriculum from smaller robots to larger robots, but the smaller robot typically faces more challenges when solving the original tasks, *e.g.*, the same obstacles could be more difficult for it. Thus, smaller robots cannot offer useful guidance for the remaining stage in the curriculum. (e-f) Our method designs robots in a coarse-to-fine manner and can focus on promising regions with the helpful guidance of coarse-grained design. Finally, our method successfully finds a simple but effective design to solve this challenging task.

them. The key insight to our approach is that the coarse-grained robot design stage can provide computationally *inexpensive* guidance for subsequent fine-grained design stages and help focus on refining promising robot designs. There are two reasons for this insight. First, coarse-grained robots are less complicated to design and control due to their low degree of freedom. Second, coarse-grained robots can usually solve part of the tasks (Figure 1 (e)), and serve as the starting points for the subsequent fine-grained design stage (Figure 1 (f)). In this way, coarse-to-fine robot design can potentially learn simple but effective robots for the given tasks. In literature, the idea of coarse-to-fine is common in computer vision (Zhu et al., 2015; Pavlakos et al., 2017) and molecule generation (Qiang et al., 2023), but has not been well studied in the field of robot design to our best knowledge.

The realization of coarse-to-fine robot design involves two major challenges. The first challenge is how to define the granularity of robots. To consider a wide range of granularities for robots of any shape, we propose to coarsen robots by aggregating adjacent cells based on their initial positions. The higher the degree of aggregation, the coarser the robots are. The coarse-grained robots under this definition are less complicated and informative for refinement because of their similarity to fine-grained robots. The second challenge is how to determine the coarse-to-fine interval, *i.e.*, when to refine current coarse-grained robots. Specifically, the design space of any-grained robots can be formulated as a tree structure, in which the parent nodes correspond to coarse-grained robots, and the child nodes represent fine-grained robots similar to their parent robots. Optimizing in this discrete tree is troublesome, as we have to determine whether to search child nodes or sibling nodes. To tackle this challenge, we introduce a novel Hyperbolic Embeddings for Robot Design (HERD) framework, where we first embed the above tree into hyperbolic space, which is naturally suitable for hierarchical representations, and then use a modified Cross-Entropy Method to optimize the robot design in this hyperbolic space. As a result, HERD unifies and optimizes any-grained robots in a shared space, and it demonstrates significant improvements over other methods.

We evaluate our HERD framework on 15 tasks in the benchmark EvoGym (Bhatia et al., 2021). HERD significantly outperforms previous state-of-the-art algorithms in terms of both sample efficiency and final performance on most tasks. Training performance comparison and visualization of optimizing in hyperbolic space strongly support the effectiveness of our method. Our experimental results highlight the importance of coarse-to-fine robot design and the representation learning of robots.

## 2 RELATED WORKS

**Multi-Cellular Soft Robot Design.** Automatic robot design focuses on generating robots that possess easy controllability for a wide range of tasks. It requires concurrently optimizing the robot morphology and control policy. A line of works in this field aims for evolving morphologies composed of rigid elements, including skeletal structures and attributes of limbs and joints (Sims, 1994a; Lipson & Pollack, 2000; Lehman & Stanley, 2011; Yuan et al., 2021). However, this setting will result in limited design space due to the compositions of predefined elements and restricted capacity of robots to interact with environments (Cheney et al., 2014b). To design robots with stronger generalization and better flexibility (Pfeifer et al., 2007), multi-cellular soft robots that are constructed by combining small cells might be a promising alternative. Previous works in multi-cellular robot design directly explore the vast design space through evolutionary search or reinforcement learning (Cheney et al., 2018; Jelisavcic et al., 2019; Walker & Hauser, 2021). Yet, evolutionary search is sample inefficient, and reinforcement learning can be easily trapped in local optimal due to exploration issues (Spielberg et al., 2019). To alleviate the burden of the thorough search, recent work CuCo incorporated curriculum learning to design processes by gradually expanding robot size (Wang et al., 2022). However, CuCo required prior knowledge to predefine the curriculum, and smaller robots might not be helpful in complex tasks. In comparison, our work enables automatic determination of coarse-to-fine transition during training, and coarse-grained robots are informative for refinement.

**Coarse-to-Fine.** Coarse-to-fine strategy refers to progressively refining a system from high-level representations by incorporating detailed information. It has been applied for some intelligent systems (Pedersoli et al., 2015) mainly including computer vision, robot manipulation, and molecule generation. Previous works on computer vision primarily mimic natural coarse-to-fine process to capture features through hierarchical localization paradigm (Sarlin et al., 2019; Yu et al., 2021), or propose a unified model architecture that enables efficient parameter reuse between different training stages (Qian et al., 2020; Dou et al., 2022). Other works in robot manipulation use this mechanism to address exploration difficulty in sparsely-rewarded and long-horizon tasks (Johns, 2021; James et al., 2022). In the field of drug discovery, it allows for 3D non-autoregressive molecule generations with a hierarchical diffusion-based model (Qiang et al., 2023). The coarse-to-fine strategy has shown its strengths in other fields. However, to our best knowledge, this strategy has not been well studied in robot design problems and our method is the first work to apply it in multi-cellular robot design.

**Reinforcement Learning for Incompatible State-Action Space.** Robot design typically requires learning generalizable control policies that could be shared among robots of diverse morphologies where the state and action spaces are incompatible. To address this challenge, prior works mainly utilize modular reinforcement learning to control each actuator separately. In the case of multiple different morphologies, one approach is to leverage GNNs to learn joint controllers (Khalil et al., 2017; Huang et al., 2020). While GNN-based methods employ graph representation relevant to the robot's morphology to ensure message passing in a shared policy (Battaglia et al., 2018; Yuan et al., 2021), Transformer-based approaches with attention mechanisms can further improve generalization, dealing with the limitation of aggregating multi-hop information (Kurin et al., 2020). In this paper, we build our control policy based on Transformer (Vaswani et al., 2017) to dynamically track the relationships between components of the robots (Gupta et al., 2021a; Dong et al., 2022).

## 3 PRELIMINARIES

In this section, we present essential background knowledge and notations for our method.

### 3.1 HYPERBOLIC SPACE AND POINCARÉ BALL

**Riemannian manifold.** A *Riemannian manifold* is defined as a tuple $(\mathcal{M}, \mathfrak{g})$ (Petersen, 2006), where $\mathcal{M}$ and $\mathfrak{g}$ are the *manifold* and the *metric tensor*, respectively, as defined below. A *manifold* $\mathcal{M}$ is a set of points $z$ that are locally similar to the linear space. Every point $z$ of the manifold $\mathcal{M}$ is attached to a *tangent space* $\mathcal{T}_z\mathcal{M}$, which is a real vector space of the same dimensionality as $\mathcal{M}$ and contains all the possible directions that can tangentially pass through $z$. Each point $z$ of the manifold also associates a *metric tensor* $\mathfrak{g}$ that defines an inner product of the tangent space: $\mathfrak{g} = \langle \cdot, \cdot \rangle_z : \mathcal{T}_z\mathcal{M} \times \mathcal{T}_z\mathcal{M} \to \mathbb{R}$. Then the inner product can induce *norm* on $\mathcal{T}_z\mathcal{M} : \| \cdot \|_z = \sqrt{\langle \cdot, \cdot \rangle_z}$.

**Hyperbolic space.** A $d$-dimensional hyperbolic space $\mathbb{H}^d$ is a $d$-dimensional Riemannian manifold with constant negative curvature $-c$. $\mathbb{H}^d$ can be represented using various isomorphic models, such as the hyperboloid model, the Beltrami-Klein model, the Poincaré half-plane model, and the Poincaré ball (Beltrami, 1868). In this paper, we use Poincaré ball for hyperbolic embeddings.

**Poincaré ball**. The Poincaré ball can be formally defined as a Riemannian manifold $\mathbb{B}_c^d = (\mathcal{B}_c^d, \mathfrak{g}_c)$, where $\mathcal{B}_c^d = \{ z \in \mathbb{R}^d : c\|z\| < 1 \}$ is an open ball, and $\mathfrak{g}_c$ is the metric tensor defined as $\mathfrak{g}_c = (\lambda_z^c)^2 \mathfrak{g}^E(z)$ where $\lambda_z^c = \frac{2}{1-c\|z\|^2}$ and $\mathfrak{g}^E = \mathbb{I}^n$ is the Euclidean metric tensor, *i.e.*, the usual dot product.

Using this metric tensor, we can induce the distance in Poincaré ball:

$$\mathcal{D}_c(z_1, z_2) = \frac{1}{\sqrt{c}} \cosh^{-1}\left(1 + 2c\frac{\|z_1 - z_2\|^2}{(1 - c\|z_1\|^2)(1 - c\|z_2\|^2)}\right). \tag{1}$$

To connect hyperbolic space and Euclidean space, we can use an exponential map and a logarithm map to map from Euclidean space to the hyperbolic space and *vice versa* with anchor $z$. On the Poincaré ball, both maps have the closed-form expressions:

$$\exp_z^c(v) = z \oplus_c \left( \tanh\left(\sqrt{c}\frac{\lambda_z^c \|v\|}{2}\right) \frac{v}{\sqrt{c}\|v\|} \right) \tag{2}$$

$$\log_z^c(v) = \frac{2}{\sqrt{c}\lambda_z^c} \tanh^{-1}\left(\sqrt{c}\| - z \oplus_c v\|\right) \frac{-z \oplus_c v}{\| - z \oplus_c v\|} \tag{3}$$

Here the operator $\oplus_c$ is the Möbius addition (Ungar, 2022) defined as

$$z \oplus_c v = \frac{(1 + 2c\langle z, v\rangle + c\|v\|^2)z + (1 - c\|z\|^2)v}{1 + 2c\langle z, v\rangle + c^2\|z\|^2\|v\|^2}. \tag{4}$$

Note that we can recover the Euclidean addition as $c \to 0$.

## 3.2 SARKAR'S CONSTRUCTION FOR HYPERBOLIC EMBEDDING

Embedding a tree into Poincaré ball $\mathbb{B}_c^d$ has many solutions. We will introduce a training-free combinatorial construction method: Sarkar's Construction (Sarkar, 2011), whose resulting embeddings can exhibit arbitrarily low distortion (Sala et al., 2018). For simplicity, we discuss the setting where dimensionality $d = 2$ and curvature $c = 1$. The basic idea is to embed the tree's root at the origin and recursively embed the children of each node by spacing them evenly around a sphere centered at the parent node as in Algorithm 2. Specifically, consider a node $q$ of degree $\deg(q)$ with parent node $p$ and suppose $q$ and $p$ have already been embedded into $z_q$ and $z_p$ in $\mathbb{B}^2$ (omitting curvature $c = 1$). We now place the children $a_1, a_2, \cdots, a_{\deg(q)-1}$ of $q$ into $\mathbb{B}^2$.

Several steps are needed. First, $z_q$ and $z_p$ are reflected across a geodesic of the Poincaré ball $\mathbb{B}^2$ by using circle inversion so that $z_q$ is mapped to the origin $\mathbf{0}$ and $z_p$ is mapped to some point $z_p'$. The reflection map is:

$$\mathcal{F}_{z_q \to \mathbf{0}}(z_p) = u + \frac{\|u\|^2 - 1}{\|z_p - u\|^2}(z_p - u) \tag{5}$$

where $u = z_q/\|z_q\|^2$. Next, place the children nodes to embeddings $y_1, y_2, \cdots, y_{\deg(q)-1}$ that are equally spaced around a circle with radius $\frac{e^\tau - 1}{e^\tau + 1}$ where $\tau$ is a scaling factor, and maximally separated from the reflected parent node embedding $z_p'$. One embedding method is:

$$y_n = \frac{e^\tau - 1}{e^\tau + 1} \cdot \left( \cos\left(\theta + \frac{2\pi n}{\deg(q)}\right), \sin\left(\theta + \frac{2\pi n}{\deg(q)}\right) \right) \tag{6}$$

where $\theta = \arg(z_p')$ is the angle of $z_p'$ from x-axis in the plane. Lastly, reflect all the points $z$ back using the same reflection map: $\mathcal{F}_{z_q \to \mathbf{0}}(z)$. To embed the entire tree, we first position the root node at the origin and its children in a circle around it, then recursively position children nodes as discussed above until all nodes are placed. Figure 2 (b) provides an intuitive example for this construction.

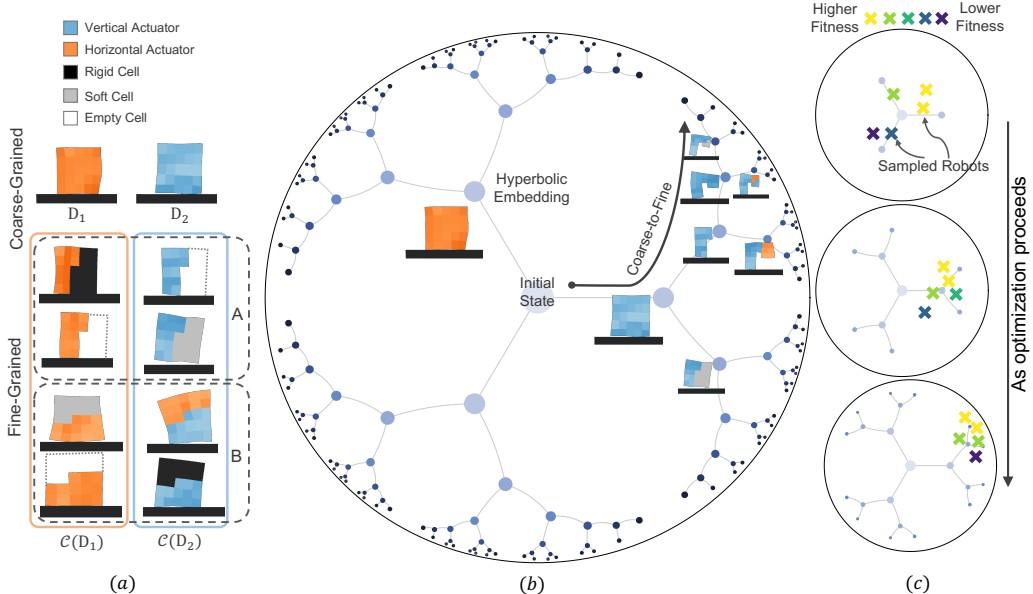

Figure 2: Hyperbolic embeddings for coarse-to-fine robot design framework. (a) An example of coarse-grained and fine-grained robots of EvoGym (Bhatia et al., 2021), where the designs $\mathcal{C}(D_i)$ are $D_i$'s child robots, *i.e.*, refined robots and the robots in A and B boxes are all fine-grained but have different cell segmentation styles; (b) Hyperbolic embeddings in the Poincaré ball for any-grained robot designs; (c) Our method is initialized at the center of the Poincaré ball, and automatically learn to move to promising regions that may close to the edge. An important property is that sampling robots from the center to the border is exactly the process of coarse-to-fine robot design.

## 4 METHOD

In this section, we present our novel learning framework that leverages Hyperbolic Embeddings for coarse-to-fine Robot Design (HERD). As in most previous works on robot design, our learning framework mainly consists of robot design optimization and control policy learning. The novelty of our framework is to design robots in a coarse-to-fine manner whose realization relies on hyperbolic embeddings. To this end, our method is characterized by two components: (1) embedding any-grained robot designs into hyperbolic space (Section 4.1) and (2) optimizing robot designs in this hyperbolic space (Section 4.2). The algorithm is outlined in Algorithm 4.

### 4.1 EMBEDDING ANY-GRAINED ROBOT DESIGNS INTO HYPERBOLIC SPACE

In this subsection, we first discuss the definition of granularity of robots in multi-cellular robot designs, then reveal the nature of the hierarchical structure of design space when learning in a coarse-to-fine manner, and finally, discuss why and how to embed any-grained robot designs in hyperbolic space. In this paper, we will focus on multi-cellular robot design and use EvoGym (Bhatia et al., 2021) as our benchmark, where each robot design $D \in \mathbb{D}$ consists of $N_c (= 25)$ inter-connected robot cells. Each robot cell has a fixed initial position $(x, y, z)$ and will be assigned a cell type chosen from {Empty, Rigid, Soft, Horizontal Actuator, Vertical Actuator} during the design process.

**Granularity** refers to the extent to which groups of small *components* of robots, *e.g.*, robot cells, have joined together to become large *components*. Coarse-grained robot designs have fewer, larger discrete components than fine-grained robot designs. The benefit of coarse-graining is that it can remove certain redundant degrees of freedom, such as the designing types and controlling actions between several neighboring robot cells.

To determine which robot cells can be aggregated together, we use a simple but effective clustering algorithm, K-Means (Hartigan & Wong, 1979), purely based on the position features of robot cells, to cluster *neighboring* robot cells into groups. The larger the number of clusters, the finer the granularity of the robots. In practice, we cluster the robot cells into $k$ components iteratively, where $k \in \{16, 8, \cdots, 1\}$. Importantly, the clustering results in the last iteration will be used as the input to

K-Means in the next iteration. Figure 2 (a) shows several results of this clustering, where robots in A and B boxes are fine-grained robots but with different clustering results due to the randomness of K-Means. It is worth noting that our method can also use other clustering methods.

**Hierarchical structure in coarse-to-fine robot design.** Given coarse-grained robots ($D_i$ in Figure 2 (a)), we can refine them by dividing large components of these robots into smaller ones ($\mathcal{C}(D_i)$ in Figure 2 (a)) according to the clustering results for fine-grained robots mentioned above. All child robots originating from the same parent robot use the same clustering results. By recursive refinement, the space of robot designs with varying granularity can be organized into a hierarchy in which the parent nodes are coarse-grained robots, and the child nodes are fine-grained robots that are similar to their parent robots. In practice, similarity here means that fine-grained robots only need to change one component to be the same as their parent robots. In the following, we use $D_0$ to represent the root node of the hierarchy and use $\mathcal{C}(D)$ to denote robot design $D$'s child robots. Denote the robot hierarchy with $\{D_i, \mathcal{C}(D_i)\}_{i=1}^N$, where $N$ is a parameter controlling the size of hierarchy.

**Why utilize hyperbolic space.** Directly searching for the optimal robot in the above robot hierarchy is cumbersome because we have to determine the coarse-to-fine interval, *i.e.*, when to refine current robots. Short intervals may lead to wrong branches of the robot hierarchy due to inaccurate performance evaluation, and long intervals may result in inefficiency. This is verified in Section 5.3.

To solve this problem, we propose to embed the robot hierarchy into a shared hyperbolic space. There are two main advantages to this solution. First, embedding any-grained robot designs in a shared space allows us to search for coarse-grained robots and refine promising ones in a unified way. This can lead to simpler and more efficient optimization, as shown in Section 5. Second, hyperbolic space is quite suitable for representing hierarchies, and actually, it can be viewed as a continuous analog of trees (Cetin et al., 2022; Hsu et al., 2021). In particular, the typical geometric property of hyperbolic space is that its volume increases exponentially in proportion to its radius, whereas the Euclidean space grows polynomically (Yang et al., 2022). Because of this nature, it is possible to embed robot hierarchy into Poincaré ball $\mathbb{B}_c^d$, one of the representations for hyperbolic space, with arbitrarily low distortion (Sarkar, 2011; Sala et al., 2018). Remarkably, Euclidean space cannot embed hierarchies with arbitrarily low distortion for *any* number of dimensions. Section 3.1 provides formal discussion.

---

**Algorithm 1:** HERD: Hyperbolic Embeddings for Coarse-to-Fine Robot Design

---

**Input:** robot design space $\mathbb{D}$, hierarchy size $N$, Poincaré ball $\mathbb{B}_c^d$, population size of CEM $N_v$

1   $\{D_i, \mathcal{C}(D_i)\}_i^N \leftarrow$ build the robot hierarchy for designs $\{D_i\}_i^N \subset \mathbb{D}$ using K-Means;
2   $\mathbb{S} = \{D_i, z_i\}_i^N \leftarrow$ embed the robot hierarchy $\{D_i, \mathcal{C}(D_i)\}_i^N$ in Poincaré ball $\mathbb{B}_c^d$ by applying Sarkar's Construction in Algorithm 2 recursively;
3   initialize control policy $\pi$, CEM mean $\boldsymbol{\mu}$ and variance $\boldsymbol{\sigma}$;
4   **while** *not reaching max iterations* **do**
5      replay buffer $\mathcal{H} \leftarrow \emptyset$;
6      **for** $i \in \{1, 2, \cdots, N_v\}$ **do**
7          $v_i \sim \mathcal{N}(\boldsymbol{\mu}, \text{diag}(\boldsymbol{\sigma}))$ ; *// sample an embedding from Euclidean space*
8          $\bar{z}_i = \exp_0^c(v_i)$ ; *// map to Poincaré ball, Equation (2)*
9          $D_i, z_i \leftarrow \arg\min_{(D, z) \in \mathbb{S}} \mathcal{D}_c(\bar{z}_i, z)$ ; *// find the nearest valid embedding and its corresponding design, Equation (1)*
10          use $\pi$ to control current robot design $D_i$ and store trajectories to $\mathcal{H}$;
11      update $\pi$ with PPO using samples in $\mathcal{H}$;
12      update $\boldsymbol{\mu}$ by averaging the elite $v_i$s based on the performance in $\mathcal{H}$, and linearly decrease $\boldsymbol{\sigma}$;
13   $D^*, z^* \leftarrow \arg\min_{(D, z) \in \mathbb{S}} \mathcal{D}_c(\exp_0^c(\boldsymbol{\mu}), z)$ ; *// optimal robot design*
14   **Output:** optimal robot design $D^*$, control policy $\pi$

---

**How to embed in hyperbolic space.** Embedding the robot hierarchy into hyperbolic space can be realized with two main strategies. The first is to use a loss function and gradient descent to learn the embedding, such as Poincaré Variational Auto-Encoder (VAE) (Mathieu et al., 2019). However, Poincaré VAE cannot embed too many randomly generated robots as we will do during the training. Thereby, we resort to the second, training-free strategy, which is termed Sarkar's Construction (Sarkar, 2011). The idea of Sarkar's Construction is simple but effective. First, it places the root node $D_0$ of the given hierarchy at the origin of Poincaré ball and places all its children nodes $\mathcal{C}(D_0)$ equally

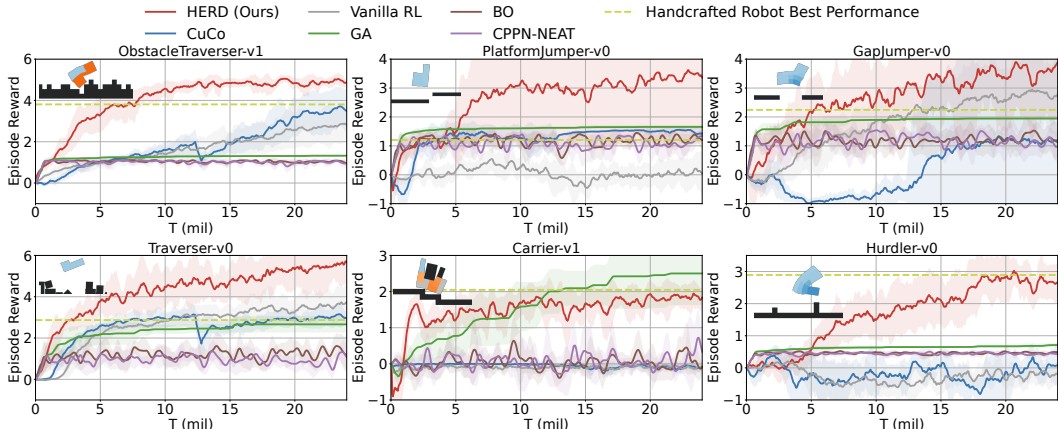

Figure 3: Training performance of HERD compared against baselines in **hard** tasks.

spaced in a circle around it. It then moves one of the children to the origin using hyperbolic reflection defined in Equation (5) and repeats. In the following, we use $\mathbb{S} = \{D_i, z_i\}_i^N$ to represent the robots and their resulting embeddings.

We show the embedding results of a simplified robot hierarchy in Figure 2 (b), where each node is the embedding for its corresponding robot design. For more details, please refer to Section 3.2. A very important property of this representation is that as the norm of the embedding increases, the robots become more fine-grained. This property allows us to realize coarse-to-fine in an elegant way.

## 4.2 OPTIMIZING ROBOT DESIGNS IN HYPERBOLIC SPACE

To optimize robot design in hyperbolic space $\mathbb{B}_c^d$, our method HERD adapts the idea of the Cross-Entropy Method (CEM) (Rubinstein & Kroese, 2004). It works by repeating the following four phases: (1) sampling embeddings from a normal distribution, (2) finding the corresponding robots of the embeddings, (3) evaluating these robots, and (4) updating the probability distribution based on the top-performing samples. Figure 2 (c) shows an example of this optimization in Poincaré ball.

Recalling that sampling robots from the center to the border of Poincaré ball is exactly the process of refining robots, we only need to initialize the mean $\boldsymbol{\mu} \in \mathbb{R}^d$ of CEM to zero to enable coarse-to-fine robot design. To control the exploration rate, we use a time-decayed variance $\boldsymbol{\sigma} \in \mathbb{R}^d$, which decreases from 0.2 to 0.01 linearly during training.

At each iteration, HERD samples $N_v$ embeddings $\{\boldsymbol{v}_i | \boldsymbol{v}_i \sim \mathcal{N}(\boldsymbol{\mu}, \text{diag}(\boldsymbol{\sigma})) \ \forall i = 1, 2, \cdots, N_v\}$. To find the corresponding robots, HERD maps each $\boldsymbol{v}_i$ from Euclidean space to the Poincaré ball by using the $\exp$ map defined in Equation (2), *i.e.*, $\bar{\boldsymbol{z}}_i = \exp_{\mathbf{0}}^c(\boldsymbol{v}_i)$. As not all points in Poincaré ball are valid under Sarkar's Construction, HERD chooses the nearest $z_i$ in the robot hierarchy with the distance function $\mathcal{D}_c$ defined in Equation (1) and uses $z_i$'s robot for $\bar{\boldsymbol{z}}_i$.

Next, the generated robots are evaluated in the given task with a shared control policy $\pi$ learned as in CuCo (Wang et al., 2022), which adopts self-attention mechanism (Vaswani et al., 2017) to handle incompatible state-action space and capture the internal dependencies between robot cells. The shared control policy is learned using PPO (Schulman et al., 2017), an actor-critic reinforcement learning algorithm.

Finally, we update the mean $\boldsymbol{\mu}$ by averaging the elite $\boldsymbol{v}_i$s based on their fitness defined by the discounted sum of rewards $\sum_t \gamma^t r_t$, where $\gamma$ is the discount factor and $r_t$ is the reward at time step $t$.

## 5 EXPERIMENTS

In this section, we benchmark our method HERD on various tasks from EvoGym (Bhatia et al., 2021). We evaluate the effectiveness of HERD by answering the following questions: (1) Can robot design help improve performance in comparison with handcrafted design? (Section 5.2) (2) Can HERD outperform other robot design baselines in various tasks? (Section 5.2) (3) How does each component of HERD contributes to its performance? (Section 5.3) (4) How does HERD design

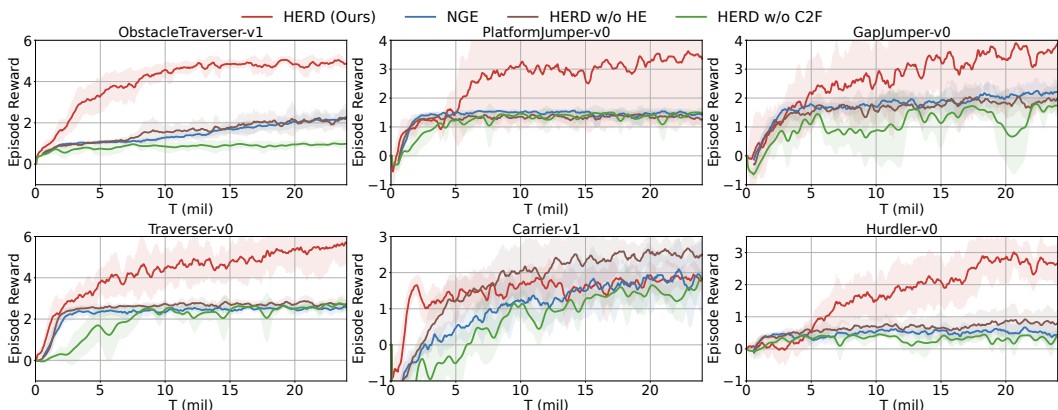

Figure 4: Ablation studies of HERD in **hard** tasks.

robots in a coarse-to-fine manner? (Section 5.4). For qualitative results, please refer to the videos on our anonymous project website[*]. And our code is available on GitHub[*].

## 5.1 EXPERIMENT SETUP

We conduct experiments on 15 tasks from EvoGym and only show the results of **hard** tasks in this section. Please refer to Appendix C for the full results. All runs are conducted with four random seeds, and the mean value as well as 95% confidence intervals are shown.

**Environments.** We follow the original setting of EvoGym, where each robot is made up of $5 \times 5$ inter-connected 2D voxel cells. Each cell has a unique type chosen from five options: empty, soft, rigid, horizontal actuator, and vertical actuator. The design space is huge with roughly $2.98 \times 10^{17}$ designs. The designed robot is controlled by assigning actions to all the *actuator* cells. The action value $a$ is within the range $[0, 6, 1.6]$ and corresponds to a gradual expansion/contraction of that actuator to $a$ times its rest length. The designed robots are required to perform multiple tasks, varying in objectives (locomotion, manipulation), terrains (flat, bumpy), and difficulties (**easy**, **medium**, **hard**). Please refer to Appendix B.1 for a detailed description of these tasks.

**Implementations and Baselines.** Our method HERD is implemented on the top of CuCo (Wang et al., 2022) by replacing its design policy with our novel optimization method in hyperbolic space. We use the same control policy and hyperparameters as CuCo for a fair comparison. To show the strengths of HERD, we consider the following baselines: (1) CuCo: a curriculum-based robot design method by expanding the design space from small size ($3 \times 3$) to the target size ($5 \times 5$) gradually through a predefined curriculum; (2) GA (Michalewicz & Michalewicz, 1996; Sims, 1994a; Wang et al., 2019): a widely used black-box optimization method by relying on biologically inspired operators such as mutation, crossover and selection. It directly encodes the robot as a vector, where each element is the cell type. (3) CPPN-NEAT (Cheney et al., 2014a;b; Corucci et al., 2018): the predominant method for evolving soft robot design. It indirectly encodes robots by Compositional Pattern Producing Network (CPPN) (Stanley, 2007; Cheney et al., 2014a) and trained by NeuroEvolution of Augmenting Topologies (NEAT) (Stanley & Miikkulainen, 2002). (4) BO (Kushner, 1964; Močkus, 1975): a commonly used global optimization method of black-box functions by training a surrogate model based on Gaussian Process (GP) to alleviate the burden of computationally expensive fitness evaluation. (5) Vanilla RL: a strong baseline that uses reinforcement learning to update the design policy and control policy. (6) HandCrafted Robot: a human-designed robot using expert knowledge that can efficiently solve many tasks. Please refer to Appendix B.2 and Appendix B.3 for more details of HERD and other baselines.

## 5.2 TRAINING PERFORMANCE COMPARISON

We summarize the training performance in Figure 3 where the x-axis and y-axis are timesteps and episode reward, respectively. We also show one representative robot designed by HERD at the end of training in the upper left corner of each sub-figure to offer more intuition, and the time-lapsed images of the control stage for these robots are provided in Appendix B.1. Here, we only show the *hard*

---

[*]https://sites.google.com/view/hyperbolic-robot-design
[*]https://github.com/drdh/HERD

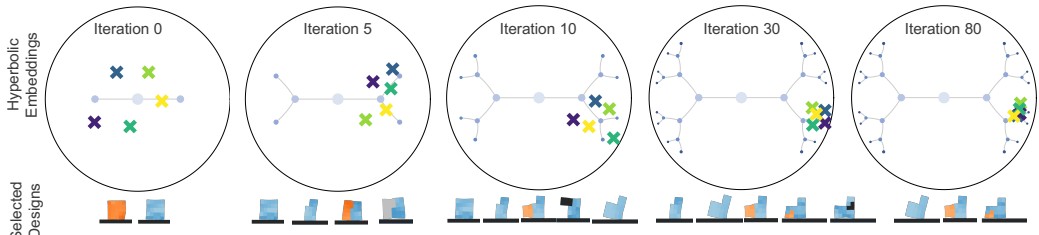

Figure 5: Visualization of the learning process of HERD in PlatformJumper-v0.

tasks, and please refer to Figure 22 for full results. HERD significantly outperforms the baselines and handcrafted robots in most tasks in terms of both design efficiency and effectiveness. This validates that (1) robot design can improve final performance compared with handcrafted robots and (2) leveraging hyperbolic embedding for coarse-to-fine learning can effectively help design robots in both efficiency and final performance.

### 5.3    ABLATIONS

To investigate the contributions of each component of HERD, we conduct elaborate ablation studies here. There are two main contributions that characterize our method: (1) the idea of coarse-to-fine robot design and (2) a realization of this idea in hyperbolic space. To validate the effectiveness of our method, we designed the following ablation studies. (1) HERD *w/o* HE: Using other method, *e.g.*, an evolutionary algorithm NGE (Wang et al., 2019), to implement the idea coarse-to-fine robot design. This ablation will show the effectiveness of hyperbolic embeddings. (2) HERD *w/o* C2F: Do not adopt coarse-to-fine robot design, and use CEM to directly search the optimal design. This ablation can verify the effectiveness of coarse-to-fine robot design. (3) NGE: Directly use NGE for multi-cellular-robot design. More details are available in Appendix B.4.

We show the results of *hard* tasks in Figure 4, and please refer to Figure 23 for other tasks. HERD is better than these variants in most tasks. Specifically, HERD outperforms HERD *w/o* C2F and NGE in all tasks, which shows the effectiveness of the idea of coarse-to-fine. Further, HERD is better than HERD *w/o* HE in most tasks, which indicates the usefulness of hyperbolic embeddings in coarse-to-fine robot design. As for our HERD being inferior to HERD *w/o* HE in Carrier-v0 and Carrier-v1, we speculate that these two tasks may only need find-grained robots, whose embeddings are close to the border of Poincaré ball. There may be a problem with bits of precision in hyperbolic embeddings (Sala et al., 2018), which is further discussed in Section 6.

### 5.4    ROBOT DESIGN PROCESS ANALYSIS

To explore how HERD can learn these robot designs, in this subsection, we visualize the evolution process of the designed robots by HERD in PlatformJumper-v0 in Figure 5. We show the hyperbolic embeddings and the selected robots from five iterations of the training. Figure 5 clearly shows that as the sampled embeddings approach the border, the designed robots change from coarse-grained to fine-grained. An interesting phenomenon is that the final robot is not the finest-grained. This is partly because coarser-grained designs are already sufficient to solve some tasks in EvoGym.

## 6    DISCUSSION

In this paper, we leverage hyperbolic embeddings for coarse-to-fine robot design. The idea of coarse-to-fine avoids directly searching in the vast design space, and hyperbolic embeddings further bring us better optimization properties. Finally, the empirical evaluations and visualizations of HERD show its superior effectiveness. However, our approach still has two limitations. First, the idea of coarse-to-fine might not be helpful in the tasks that only the finest-grained robots can solve. In these tasks, coarse-grained robot designs will not provide useful guidance for refinement if the coarse-grained robots cannot address even part of the tasks. Second, embedding robots in hyperbolic space may face the problem of bits of precision (Sala et al., 2018). To embed finer-grained robots better, our method requires more bits. Specifically, given the height of the robot hierarchy $H_{\max}$, and the maximum number of child robots $\deg_{\max}$, our embedding will need $\Omega(H_{\max} \log(\deg_{\max}))$ bits (Sala et al., 2018). Solving these two limitations could be promising for future work and may further improve performance in other tasks.

## REPRODUCIBILITY STATEMENT

Our research findings are fully reproducible. The source code is included in the Supplementary Material and will be made public once accepted. We have also thoroughly documented the installation instructions and experimental procedures in a README file attached to the code.

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

## A   METHOD DETAILS

We outline the concrete algorithm for Sarkar's Construction in Algorithm 2.

---

**Algorithm 2:** Sarkar's Construction adapted from Sala et al. (2018)

---

**Input:** Node $q$ with parent $p$, children to place $a_1, a_2, \cdots, a_{\deg(q)-1}$, scaling factor $\tau$.

1 $(0, z_p') \leftarrow \mathcal{F}_{z_q \to 0}(z_q, z_p)$; ;                 // defined in Equation (5)

2 $\theta \leftarrow \arg(z_p')$; ;                      // angle of $z_p'$ from x-axis in the plane

3 **for** $n \in \{1, \cdots, \deg(q) - 1\}$ **do**

4 $\quad \Big|\quad y_n \leftarrow \frac{e^\tau - 1}{e^\tau + 1} \cdot \left( \cos\left(\theta + \frac{2\pi n}{\deg(q)}\right), \sin\left(\theta + \frac{2\pi n}{\deg(q)}\right) \right)$;

5 $(z_q, z_p, z_1, \cdots, z_{\deg(q)-1}) \leftarrow \mathcal{F}_{z_q \to 0}(0, z_p', y_1, \cdots, y_{\deg(q)-1})$;

**Output:** Embeddings in $\mathbb{B}^2$: $z_1, z_2, \cdots, z_{\deg(q)-1}$.

---

## B   EXPERIMENT DETAILS

### B.1   DETAILS OF THE TASKS

We provide details for 15 various tasks ranging from simple to complex simulated in the EvoGym platform in the following section. More elaborate explanations can be referred to Bhatia et al. (2021). In each task, the initial design space is $5 \times 5$.

**Position.** $p^o$ represents a two-dimensional position vector for the centroid of an object $o$ at time $t$, which is computed by averaging all the positions of point masses used to compose object $o$. $p_x^o$ and $p_y^o$ are $x$ and $y$ components of this vector, respectively.

**Velocity.** Likewise, $v^o$ denotes a two-dimensional velocity vector for the center of mass of an object $o$ at time $t$, which is calculated by averaging all the velocities of point-masses used to make up object $o$. $v_x^o$ and $v_y^o$ are $x$ and $y$ components of this vector, respectively.

**Orientation.** Similarly, $\theta^o$ refers to a one-dimensional orientation vector for an object $o$ at time $t$. Given $p_i$, which represents the position of point mass $i$ of object $o$ and $p^o$, we can calculate $\theta^o$ by averaging over the angle between the vector $p_i - p^o$ at time $t$ and time 0 for all the point masses. Note that this average is a weighted average determined by $||p_i - p^o||$ at time 0.

**Special observations.** $c^o$ describes a $2n$-dim position vector for all $n$ point masses of an object $o$ relative to $p^o$, which can be computed by subtracting $p^o$ from each column of the $2 \times n$ position matrix.

$h_b^o(d)$ and $h_a^o(d)$ are vectors of length $2d + 1$ that represent elevation information around the robot below or above its centroid. More specifically, for some integer $x \le d$, the corresponding entry in vector $h_b^o(d)$ will be the highest point of the terrain, which is less than $p_y^o$ between a range of $[x, x+1]$ voxels from $p_x^o$ in the x-direction.

### B.1.1   WALKER-V0

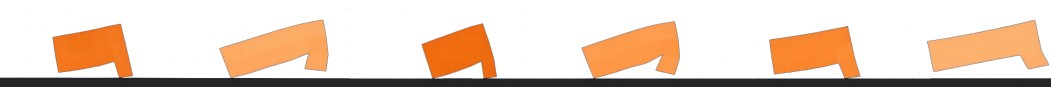

Figure 6: Walker-v0.

In this task, the robot is required to walk as far as possible on flat terrain. This task is **easy**. Given the observation space $\mathcal{S} \in \mathbb{R}^{n+2}$ that is formed by concatenating vectors $v^{robot}, c^{robot}$, where $n$ is the

number of point masses, the reward $R$ can be represented as below:

$$R = \Delta p_x^{robot}$$

which provides a reward to the robot for making forward progress in the positive x-direction.

It also receives an additional reward of 1 if the robot reaches the end of the terrain.

### B.1.2 DOWNSTEPPER-V0

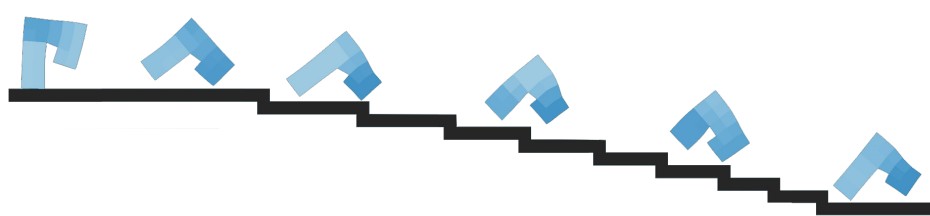

Figure 7: DownStepper-v0.

In this task, the robot achieves rewards by descending stairs of various lengths. This task is **easy**. Given the observation space $\mathcal{S} \in \mathbb{R}^{n+14}$ that is constituted by vectors $v^{robot}$, $\theta^{robot}$, $c^{robot}$, $h_b^{robot}(5)$, where $n$ is the number of point masses, , the reward $R$ can be formulated as follow:

$$R = \Delta p_x^{robot}$$

which rewards the robot for moving in the positive x-direction.

The robot also receives an additional reward of 2 upon successfully reaching the end of the terrain but will be penalized with a one-time deduction of 3 for rotating more than 75 degrees in either direction from its initial orientation, similar to the Up Stepper task.

### B.1.3 JUMPER-V0

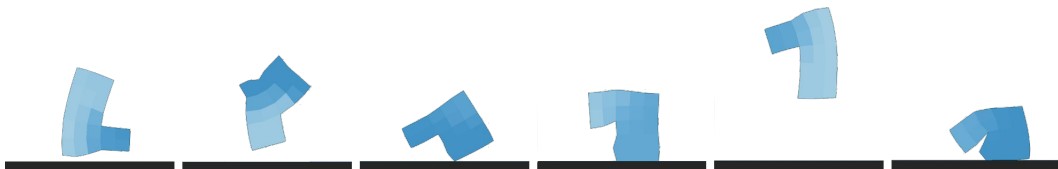

Figure 8: Jumper-v0.

In this task, the robot is required to jump as high as possible in place on flat terrain. This task is **easy**. Given the observation space $\mathcal{S} \in \mathbb{R}^{n+7}$ that is formed by concatenating vectors $v^{robot}$, $c^{robot}$, $h_b^{robot}(2)$, where $n$ is the number of point masses, the reward $R$ is

$$R = 10 \cdot \Delta p_y^{robot} - 5 \cdot |p_x^{robot}|$$

which implies that the robot will receive a reward for its upward motion in the positive y-direction and a penalty for any movement in the x-direction.

### B.1.4 FLIPPER-V0

In this task, the robot aims to perform as many counter-clockwise flips as possible on flat terrain. This task is **easy**. Given the observation space $\mathcal{S} \in \mathbb{R}^{n+1}$ that is formed by concatenating vectors $\theta^{robot}$, $c^{robot}$, the reward $R$ is represented as below:

$$R = \Delta \theta^{robot}$$

which rewards the robot for executing counter-clockwise rotations.

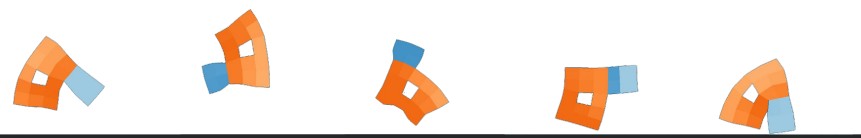

Figure 9: Flipper-v0.

### B.1.5 PUSHER-V0

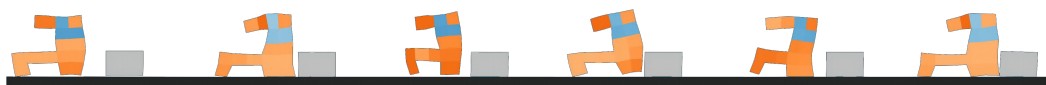

Figure 10: Pusher-v0.

In this task, the objective of the robot is to push a box that has been placed in front of it. This task is **easy**. Given the observation space $\mathcal{S} \in \mathbb{R}^{n+6}$ that is constituted by vectors $v^{box}$, $p^{box} - p^{robot}$, $v^{robot}$, $c^{robot}$, where $n$ is the number of point masses, the reward $R$ can be represented as the sum of $R_1$ and $R_2$.

$$R_1 = 0.5 \cdot \Delta p_x^{robot} + 0.75 \cdot \Delta p_x^{box}$$

which rewards the robot for its movement in the positive x-direction.

$$R_2 = -\Delta |p^{box} - p^{robot}|$$

which penalizes the robot for moving apart in the x-direction.

It also receives an additional reward of 1 if the robot reaches the end of the terrain.

### B.1.6 CARRIER-V0

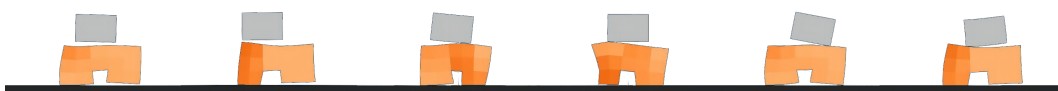

Figure 11: Carrier-v0.

In this task, the robot is required to catch a box initialized above it and carry it as far as possible on flat terrain. This task is **easy**. Given the observation space $\mathcal{S} \in \mathbb{R}^{n+6}$ that is constituted by vectors $v^{box}$, $p^{box} - p^{robot}$, $v^{robot}$, $c^{robot}$, where $n$ is the number of point masses, the reward $R = R_1 + R_2$ is the sum of several components.

$$R_1 = 0.5 \cdot \Delta p_x^{robot} + 0.5 \cdot \Delta p_x^{box}$$

which rewards both the robot and box for movements in the positive x-direction.

$$R_2 = \begin{cases} 0 & \text{if } p_y^{box} \geq t_y \\ 10 \cdot \Delta p_y^{box} & \text{otherwise} \end{cases}$$

which implies that the robot will receive a penalty for dropping the box below a threshold height $t_y$.

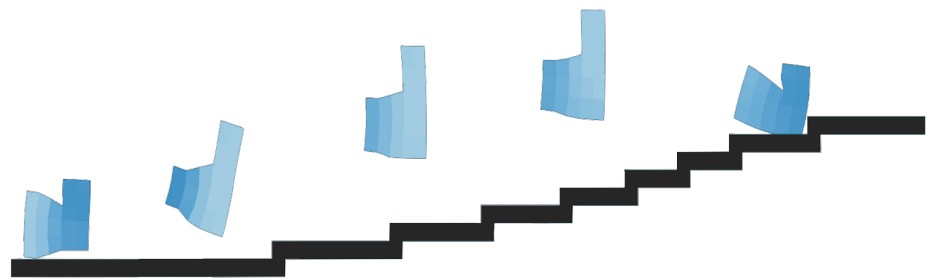

Figure 12: UpStepper-v0.

### B.1.7 UPSTEPPER-V0

In this task, the robot is rewarded by climbing up stairs of diverse lengths. This task is **medium**. Given the observation space $\mathcal{S} \in \mathbb{R}^{n+14}$ that is constituted by vectors $v^{robot}$, $\theta^{robot}$, $c^{robot}$, $h_b^{robot}(5)$, where $n$ is the number of point masses, the reward $R$ is formulated as follow:

$$R = \Delta p_x^{robot}$$

which incentivizes the robot to move in the positive x-direction.

The robot is also granted an additional reward of 2 upon reaching the end of the terrain but will be penalized with a one-time deduction of 3 for rotating more than 75 degrees in either direction from its original orientation.

### B.1.8 OBSTACLETRAVERSER-V0

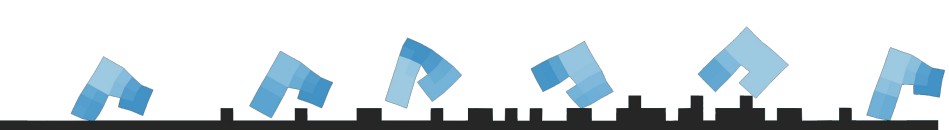

Figure 13: ObstacleTraverser-v0.

In this task, the robot is rewarded by walking across terrain that gets increasingly bumpy. This task is **medium**. Given the observation space $\mathcal{S} \in \mathbb{R}^{n+14}$ that is constituted by vectors $v^{robot}$, $\theta^{robot}$, $c^{robot}$, $h_b^{robot}(5)$, where $n$ refers to the number of point masses, the reward $R$ is formulated as follow:

$$R = \Delta p_x^{robot}$$

which incentivizes the robot to move in the positive x-direction.

The robot also receives an additional reward of 2 upon successfully reaching the end of the terrain but will be penalized with a one-time deduction of 3 for rotating more than 90 degrees in either direction from its initial orientation.

### B.1.9 THROWER-V0

In this task, the robot is rewarded by throwing a box that is initially positioned on top of it. This task is **medium**. Given the observation space $\mathcal{S} \in \mathbb{R}^{n+6}$ that is constituted by vectors $v^{box}$, $p^{box} - p^{robot}$, $v^{robot}$, $c^{robot}$, where $n$ is the number of point masses, the reward $R$ can be represented as the sum of $R_1$ and $R_2$.

$$R_1 = \Delta p_x^{box}$$

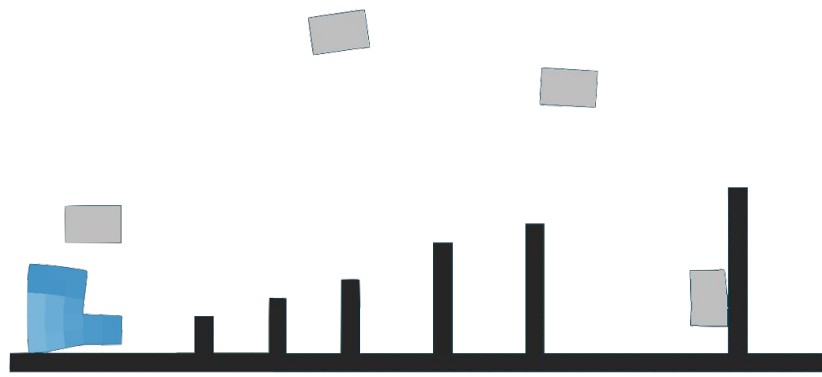

Figure 14: Thrower-v0.

which rewards the box for moving in the positive x-direction.

$$R_2 = \begin{cases} -\Delta p_x^{robot} & \text{if } p_x^{robot} \geq 0 \\ \Delta p_x^{robot} & \text{otherwise} \end{cases}$$

which imposes a penalty on the robot if it moves too far away from the $x = 0$ while throwing the box.

### B.1.10 PLATFORMJUMPER-V0

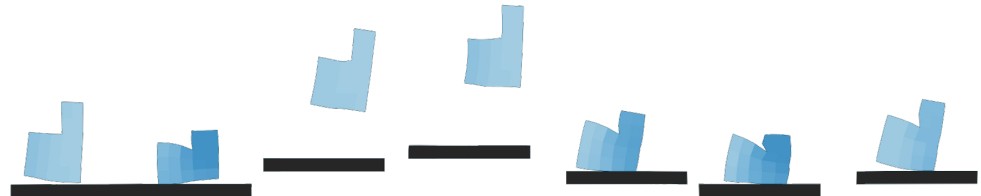

Figure 15: PlatformJumper-v0.

In this task, the robot navigates through a sequence of floating platforms positioned at varying heights. This task is **hard**. Given the observation space $\mathcal{S} \in \mathbb{R}^{n+14}$ that is constituted by vectors $v^{robot}$, $\theta^{robot}$, $c^{robot}$, $h_b^{robot}(5)$, where $n$ is the number of point masses, the reward $R$ is formulated as follow:

$$R = \Delta p_x^{robot}$$

which provides a reward to the robot for making forward progress in the positive x-direction.

If the robot rotates more than 90 degrees from its initial orientation in either direction or falls off the platforms, it incurs a one-time penalty of -3.

### B.1.11 OBSTACLETRAVERSER-V1

In this task, the robot is required to traverse a rougher terrain compared to the previous Obstacle Traverser task. This task is **hard**. Given the observation space $\mathcal{S} \in \mathbb{R}^{n+14}$ that is constituted by vectors $v^{robot}$, $\theta^{robot}$, $c^{robot}$, $h_b^{robot}(5)$, where $n$ refers to the number of point masses, the reward $R$ can be formulated as follow:

$$R = \Delta p_x^{robot}$$

which rewards the robot for its movement in the positive x-direction

The robot also receives an additional reward of 2 upon successfully reaching the end of the terrain but won't be penalized.

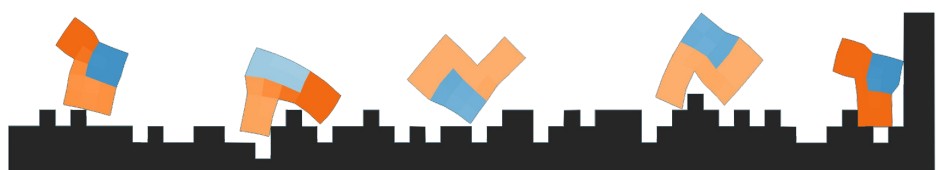

Figure 16: ObstacleTraverser-v1.

### B.1.12 HURDLER-V0

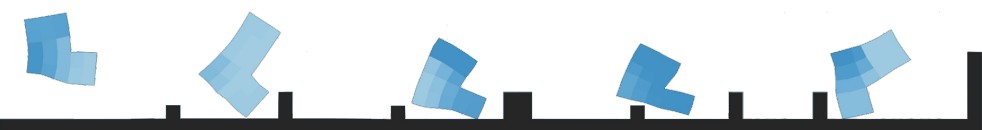

Figure 17: Hurdler-v0.

In this task, the robot achieves rewards by walking across terrain that features tall obstacles. This task is **hard**. Given the observation space $\mathcal{S} \in \mathbb{R}^{n+14}$ that is similar to ObstacleTraverser-v1, the task-specific reward $R$ can be formulated as follow:

$$R = \Delta p_x^{robot}$$

which rewards the robot for moving in the positive x-direction.

It incurs a one-time penalty of -3 if the robot deviates more than 90 degrees from its original orientation in either direction.

### B.1.13 TRAVERSER-V0

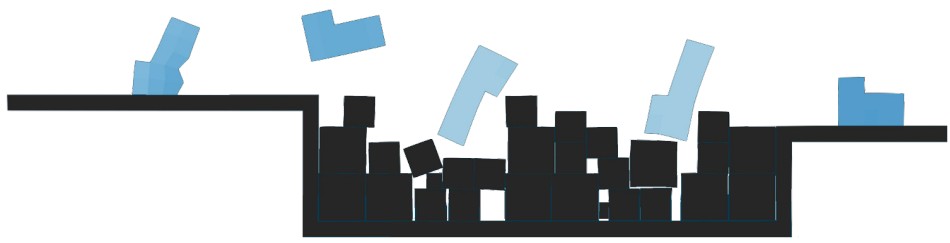

Figure 18: Traverser-v0.

In this task, the robot navigates across a pit filled with rigid blocks without sinking into the pit. This task is **hard**. Given the observation space $\mathcal{S} \in \mathbb{R}^{n+14}$ that is similar to ObstacleTraverser-v1, the task-specific reward $R$ can be represented as below:

$$R = \Delta p_x^{robot}$$

which provides a reward to the robot for making forward progress in the positive x-direction.

The robot is also granted an additional reward of 2 upon reaching the end of the terrain.

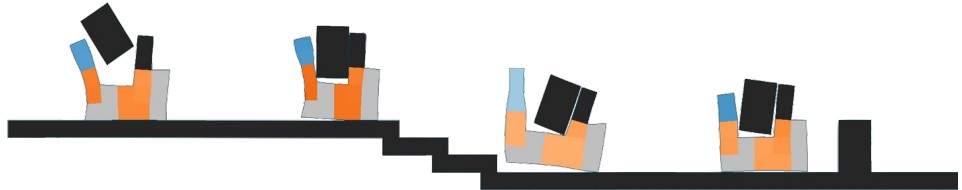

Figure 19: Carrier-v1.

### B.1.14 CARRIER-V1

In this task, the objective of the robot is to carry a box to a table and place it on the table. This task is **hard**. Given the observation space $\mathcal{S} \in \mathbb{R}^{n+6}$ that is similar to Thrower-v0, the reward $R$ can be formulated as the sum of $R_1$, $R_2$ and $R_3$.

$$R_1 = -2 \cdot \Delta |g_x^{box} - p_x^{box}|$$

where $g_x^{box}$ refers to the goal x-position for the box. It rewards the box for moving to its goal in the x-direction.

$$R_2 = -\Delta |g_x^{robot} - p_x^{robot}|$$

where $g_x^{robot}$ describes the goal x-position for the robot. The robot is rewarded for making forward progress to its target in the positive x-direction.

$$R_3 = \begin{cases} 0 & \text{if } p_y^{box} \geq t_y \\ 10 \cdot \Delta p_y^{box} & \text{otherwise} \end{cases}$$

which imposes a penalty on the robot if it drops the box below a threshold height $t_y$ that varies with the elevation of the terrain.

### B.1.15 GAPJUMPER-V0

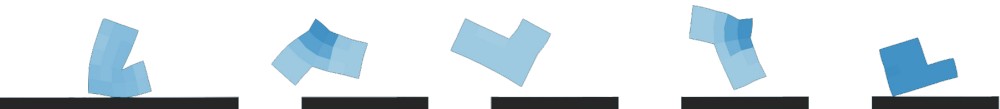

Figure 20: GapJumper-v0.

In this task, the robot moves across a sequence of spaced-out floating platforms, all positioned at the same height. This task is **hard**. Given the observation space $\mathcal{S} \in \mathbb{R}^{n+14}$ that is constituted by vectors $v^{robot}$, $\theta^{robot}$, $c^{robot}$, $h_b^{robot}(5)$, where $n$ refers to the number of point masses, the reward $R$ is formulated as follow:

$$R = \Delta p_x^{robot}$$

which incentivizes the robot to move in the positive x-direction.

If the robot falls off the platforms, it incurs a one-time penalty of -3.

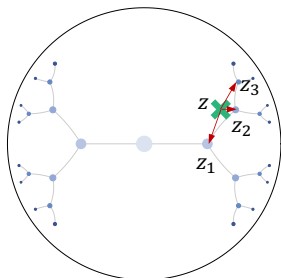

Figure 21: Use the nearest valid embedding and its corresponding design.

## B.2 IMPLEMENTATION OF OUR METHOD HERD

We implement HERD based on the public code[*] of CuCo (Wang et al., 2022), which uses Transformer-based Kurin et al. (2020) control policies. This architecture can deal with variable input sizes across different robot designs and can capture internal; dependencies between cells. Note that our method is general and can also use any other networks that are able to handle incompatible state-action space, such as GNN (Scarselli et al., 2008; Bruna et al., 2013; Kipf & Welling, 2016) and message passing network (Huang et al., 2020). For control policy learning, we use Proximal Policy Optimization (PPO) (Schulman et al., 2017) as in CuCo.

For the designing component, the implementations of Sarkar's Construction and Poincaré ball are based on the released code[*] of HoroPCA (Chami et al., 2021). In Figure 21, we use a simplified Poincaré disk, where the depth of the tree and the number of branches is reduced for better visualization, to show how we select the nearest valid embedding and its corresponding design. We simply calculate the distance between the sampled point $z$ and other points (such as $z_1, z_2, z_3, \cdots$) that are embedded in the hyperbolic manifold. Then we choose the point and its corresponding robot design with minimum distance, *i.e.*, $z_2$ in Figure 21.

Here we provide the hyperparameters needed to replicate our experiments in Table 1, and we also include our codes in the supplementary.

Experiments are all conducted on NVIDIA GTX 2080Ti GPUs with 80 CPUs. Taking ObstacleTraverser-v0 as an example, HERD requires approximately 4G of RAM and 1G of video memory and takes about 28 hours to finish 25M timesteps of training.

## B.3 DETAILS OF BASELINES

**CuCo.** CuCo (Wang et al., 2022) incorporates curriculum learning for co-optimization of design and control by establishing a simple-to-complex process. It enables gradually expanding the design space from initial size to target size through a predefined curriculum. By utilizing an indirect-encoding Neural Cellular Automata (NCA) and an inheritance mechanism, CuCo achieves a balanced exploration and exploitation. At each stage of the curriculum, it learns transferable control patterns via Reinforcement Learning (RL) and adopts the self-attention mechanism to capture the internal dependencies between voxels effectively.

**GA.** We have implemented a basic Genetic Algorithm (GA) that employs elitism selection and a simple mutation strategy to evolve the population of robot designs. The elitism selection mechanism preserves robots with high fitness from the current population as survivors while discarding the rest. The mutation strategy introduces random modifications to individual voxels of the robot with a specified probability. Although we have not incorporated crossover for the sake of simplicity, a well-designed crossover operator could potentially enhance final performance.

**CPPN-NEAT.** In this approach, the morphology of the soft robot is represented and parameterized using a Compositional Pattern Producing Network (CPPN) (Cheney et al., 2014a; Stanley, 2007).

---

[*]`https://github.com/Yuxing-Wang-THU/ModularEvoGym`
[*]`https://github.com/HazyResearch/HoroPCA`

Table 1: Hyperparameters of HERD.

| | Hyperparameter | Value |
|---|---|---|
| Design Optimization | Dimension of Poincaré ball $d$ | 2 |
| | Curvature of Poincaré ball $c$ | 1.0 |
| | Hierarchy size $N$ | $\approx 2 \times 10^4$ |
| | Population size of CEM $N_v$ | 10 |
| PPO | Use GAE | True |
| | GAE parameter $\lambda$ | 0.95 |
| | Learning rate | $2.5 \times 10^{-4}$ |
| | Use linear learning rate decay | True |
| | Clip parameter | 0.1 |
| | Value loss coefficient | 0.5 |
| | Entropy coefficient | 0.01 |
| | Time steps per rollout | 2048 |
| | Num processes | 4 |
| | Optimizer | Adam |
| | Evaluation interval | 10 |
| | Discount factor $\gamma$ | 0.99 |
| | Clipped value function | True |
| | Observation normalization | True |
| | Observation clipping | $[-10, 10]$ |
| | Reward normalization | True |
| | Reward clipping | $[-10, 10]$ |
| | Policy epochs | 8 |
| Transformer | Number of layers | 2 |
| | Number of attention heads | 1 |
| | Embedding dimension | 64 |
| | Feedforward dimension | 128 |
| | Non linearity function | ReLU |
| | Dropout | 0.0 |

As CPPN receives all the spatial coordinates of voxels as input, we can acquire the type of each corresponding voxel. The NeuroEvolution of Augmenting Topologies (NEAT) algorithm (Stanley & Miikkulainen, 2002) is leveraged to evolve the structure of CPPNs, which operates as a generic algorithm tailored for network structure.

**BO.** Bayesian Optimization (BO) (Kushner, 1964; Schaff et al., 2019) is a widely adopted technique for global optimization, especially for optimizing functions that are computationally expensive to evaluate. Specifically, the surrogate model employed in this approach is based on Gaussian processes, while batch Thompson sampling is to extract the acquisition function. We rely on the L-BFGS algorithm during the optimization process.

**Vanilla RL.** Vanilla RL method shares the same network architectures as CuCo (Wang et al., 2022), but removes the curriculum component from the framework. It learns to design and control from scratch within the design space.

**HandCrafted Robot.** To evaluate the effectiveness of automatic robot design, we construct a robot in the design phase using human knowledge, and it solely requires learning control policies for specific morphologies.

The baselines CuCo, Vanilla RL, and HandCrafted Robot adopt the same control policy architecture (transformers) and learning method (PPO) as our method HERD. For the other baselines GA, BO, and CPPN-NEAT, we use their official implementations from CuCo and EvoGym. Specifically, they also use transformers as control policy architecture and use PPO as optimization method. The main difference is that the control policy is not shared across robot designs and needs to be learned from scratch.

### B.4 DETAILS OF ABLATIONS

**HERD *w/o* HE.** The idea of coarse-to-fine can be implemented by other algorithms. HERD *w/o* HE uses NGE (Wang et al., 2019), an evolutionary algorithm originally used for rigid robot design. We re-implemented the algorithm in EvoGym based on their released code[*]. The core modification is the *mutation procedure*. To enable multi-cellular robot design, we randomly change the type of each robot cell with probability 0.1, where the types are chosen from {Empty, Rigid, Soft, Horizontal Actuator, Vertical Actuator}. To enable coarse-to-fine robot design, we refine each robot with probability 0.2 if it is coarse-grained. Then the mutated robots are viewed as offsprings and included in the robot population of NGE.

**NGE.** For full comparison, we also use the original NGE without coarse-to-fine. The modification of NGE is similar to HERD *w/o* HE and the only difference is that NGE does not have coarse-to-fine component.

**HERD *w/o* C2F.** The optimization of the robot designs in HERD is based on CEM, which works in Poincaré ball. HERD *w/o* C2F does not adopt the idea of coarse-to-fine, but uses CEM to directly search the optimal design. This ablation can verify the effectiveness of coarse-to-fine robot design. Specifically, the major difficulty of using CEM for optimizing robots in EvoGym is that the design space of EvoGym is discrete. We choose to let CEM optimize the probabilities of robot cells types ($P \in \mathbb{R}^{25 \times 5}$), and take the types with maximum probabilities ('P.argmax(dim=1)').

For a fair comparison, the control policies of all the variants in ablation studies adopt the same network architecture and same optimization algorithm (PPO) as our method HERD.

---

**Algorithm 3:** HERD *w/o* HE

---

**Input:** robot design space $\mathbb{D}$, population size of NGE $N_{nge}$= 64,

1  initialize control policy $\pi$, a population of coarse-grained robots $\{D_i\}_i^{N_{nge}}$;
2  **while** *not reaching max iterations* **do**
3       replay buffer $\mathcal{H} \leftarrow \emptyset$;
4       **for** $i \in \{1, 2, \cdots, N_{nge}\}$ **do**
5           use $\pi$ to control current robot design $D_i$ and store trajectories to $\mathcal{H}$;
6       Eliminate 30% of the robots with poor performance;
7       **while** *not reaching the population size* $N_{nge}$ **do**
8           sample one robot from the remaining robots and draw $\mu$ from $[0, 1]$ uniformly;
9           **if** $\mu < 0.8$ **then**
10             alter the type of each component with the probability of 0.1;
11          **else**
12             alter the granularity of one component ;      // coarse-to-fine mutation
13          add the offspring into the candidate pool;
14      update $\pi$ with PPO using samples in $\mathcal{H}$;
15 $D^* \leftarrow$ robot design with the highest performance ;      // optimal robot design
16 **Output:** optimal robot design $D^*$, control policy $\pi$

---

## C  EXTRA EXPERIMENTS

In this section, we provide the full experiment results of 15 tasks. Figure 22 show the training performance of HERD compared with baselines. HERD outperforms other baselines in most tasks, including **easy**, **medium** and **hard**, which further showcase the effectiveness of our method.

Figure 23 presents the performance of several variants of HERD. It again supports our observation that the idea of coarse-to-fine is helpful for robot design problems and hyperbolic embeddings can further bring better optimization properties.

The performance of HandCrafted Robot is obtained by averaging the final performance of four random runs. The reason why we did not show its learning curve in the main text is that it is not a robot design method. HandCrafted Robot has a fixed human-designed robot morphology and only

---

[*]https://github.com/WilsonWangTHU/neural_graph_evolution

---

**Algorithm 4:** HERD *w/o* C2F

**Input:** robot design space $\mathbb{D}$, population size of CEM $N_v$

1   initialize control policy $\pi$, CEM mean $\boldsymbol{\mu}$ and variance $\boldsymbol{\sigma}$;

2   **while** *not reaching max iterations* **do**

3      replay buffer $\mathcal{H} \leftarrow \emptyset$;

4      **for** $i \in \{1, 2, \cdots, N_v\}$ **do**

5          $\boldsymbol{v}_i \sim \mathcal{N}(\boldsymbol{\mu}, \text{diag}(\boldsymbol{\sigma}))$; // `sample an embedding from Euclidean space`

6          $D_i \leftarrow \arg\max_{dim=1} \boldsymbol{v}_i$;

7          use $\pi$ to control current robot design $D_i$ and store trajectories to $\mathcal{H}$;

8      update $\pi$ with PPO using samples in $\mathcal{H}$;

9      update $\boldsymbol{\mu}$ by averaging the elite $\boldsymbol{v}_i$s based on the performance in $\mathcal{H}$, and linearly decrease $\boldsymbol{\sigma}$;

10   $D^* \leftarrow \arg\max_{dim=1} \boldsymbol{\mu}$;                    // `optimal robot design`

11   **Output:** optimal robot design $D^*$, control policy $\pi$

---

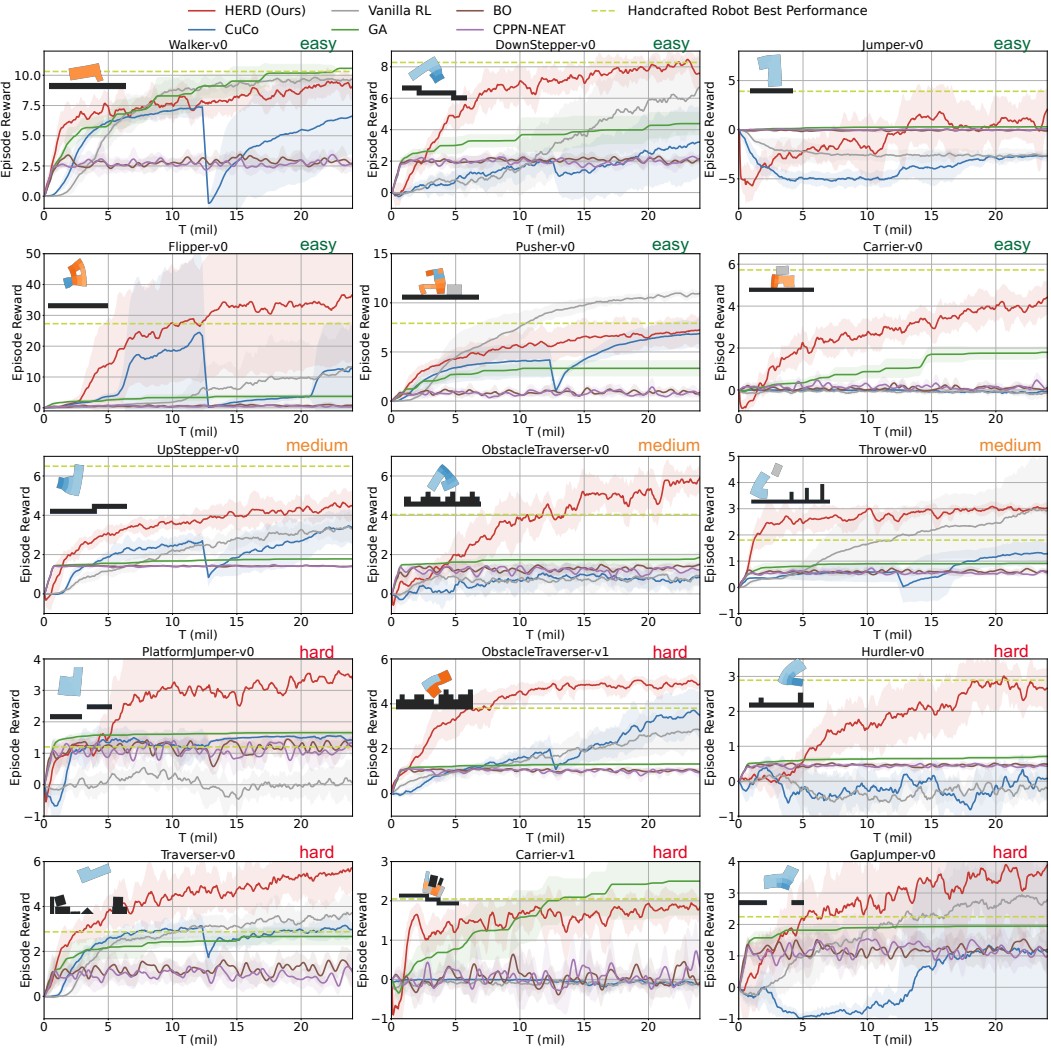

Figure 22: Training performance of HERD compared against baselines in 15 tasks.

needs to learn control policy. It is unfair to directly compare this baseline with other robot design methods, whose robot morphologies are learned online. In this paper, we use the final performance

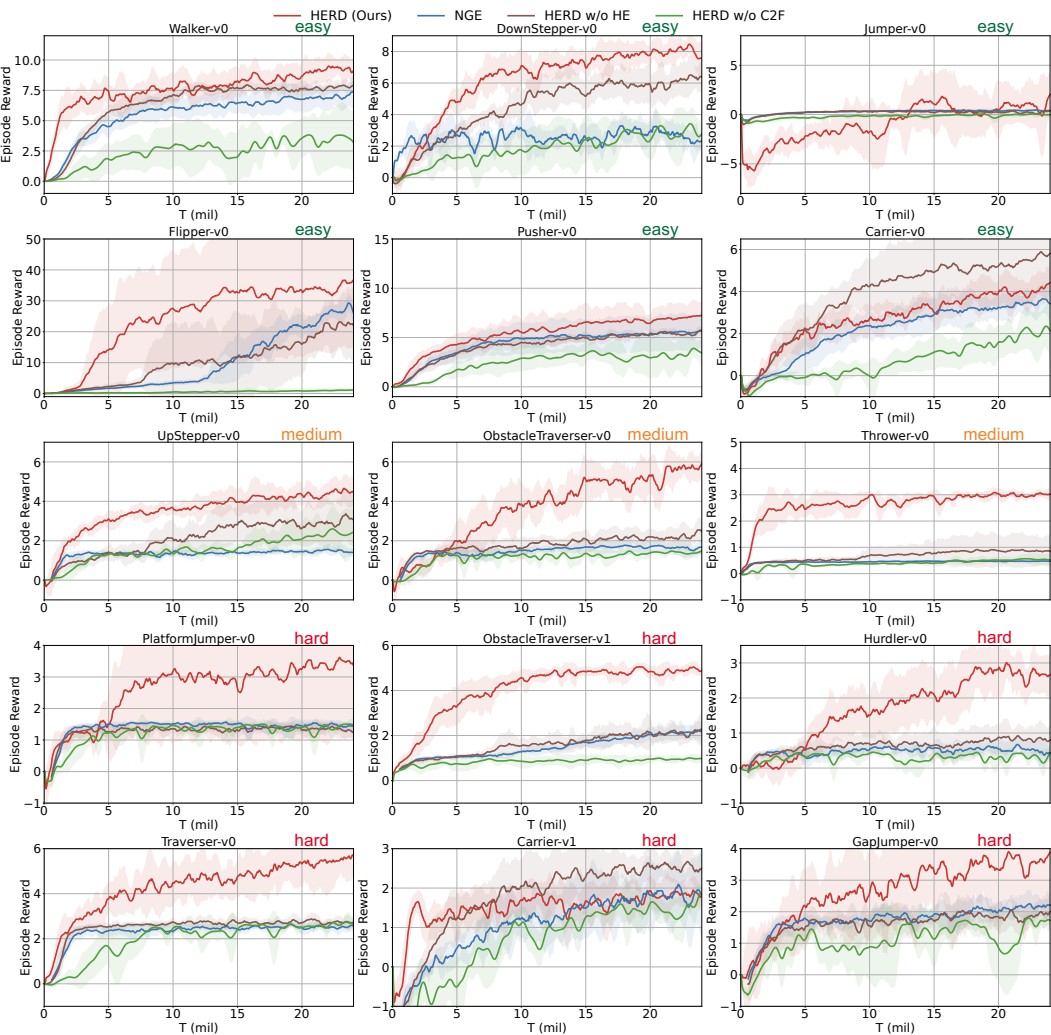

Figure 23: Ablation studies of HERD in 15 tasks.

improvement of our method over this baseline to show the necessity of creating robot design methods. However, for a complete comparison, we also include the learning curve in Figure 24.

For a full comparison, we also test our method as well as baselines in other tasks of EvoGym in Figure 25. The tasks that have not been solved by our method and other baselines can be roughly classified into two categories: (1) the tasks require complicated control policies such as Pusher-v1 and (2) the tasks require delicate robots such as Lifter-v0. These unsolved tasks might require further investigation. An we do not include the task BidirectionalWalker-v0, because it contains bugs in its task implementation.

## D QUANTITATIVE RESULTS

To better understand the evaluation of different methods, we also show the quantitative results in Table 2 and Table 3.

## E DISCUSSION OF OTHER RELATED WORK

Our work adopts an embedding approach in hyperbolic space and optimizes robot designs in this space. The most similar previous work to our paper is GLSO Hu et al. (2023), which uses a variational

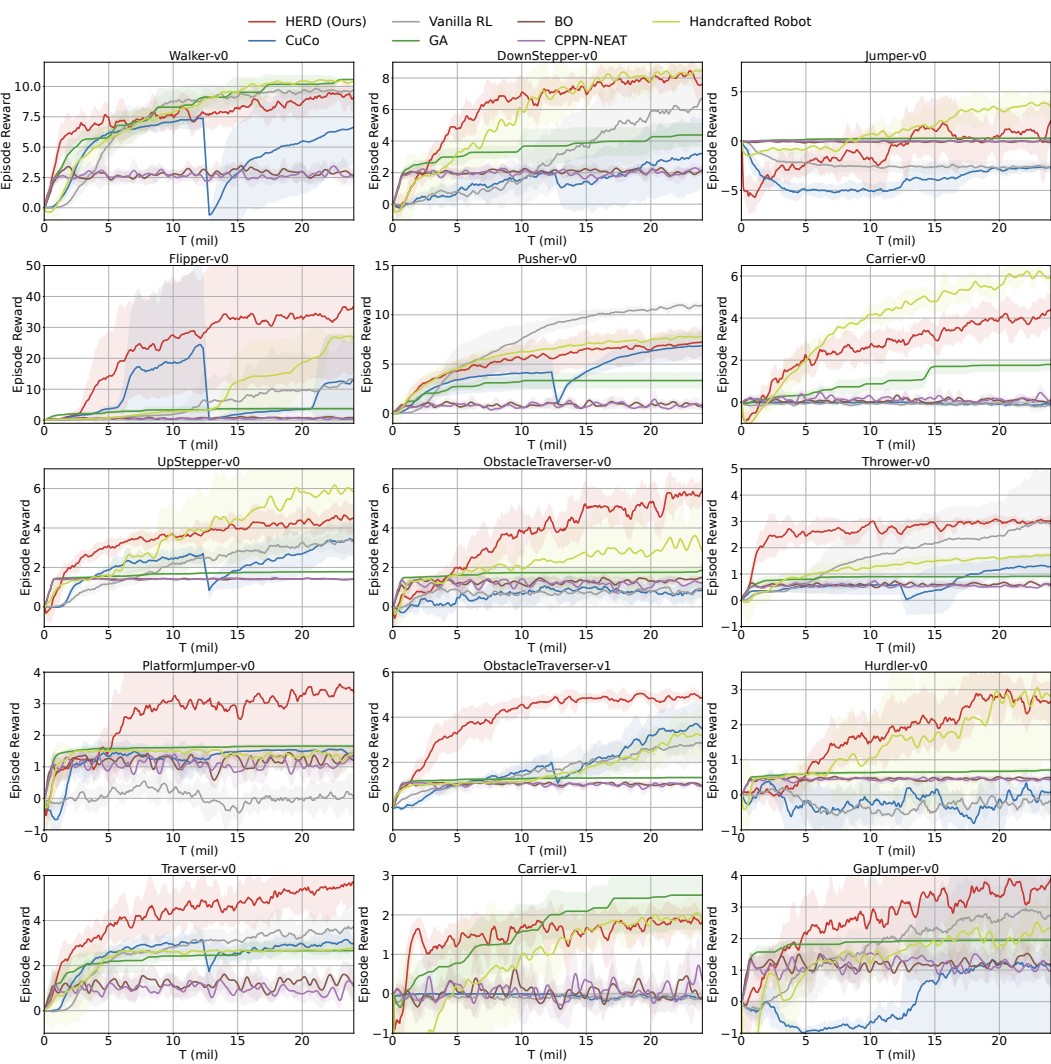

Figure 24: Training performance of HERD compared against baselines in 15 tasks.

Table 2: Quantitative results of our method HERD compared with various baselines.

|  | **HERD (Ours)** | **CuCo** | **Vanilla RL** | **GA** | **BO** | **CPPN-NEAT** | **Handcrafted** |
|---|---|---|---|---|---|---|---|
| Walker-v0 | 9.74 ± 0.81 | 6.68 ± 2.73 | 9.49 ± 0.62 | **10.57 ± 0.00** | 3.13 ± 0.73 | 2.22 ± 0.67 | **10.50 ± 0.10** |
| DownStepper-v0 | **8.04 ± 0.76** | 3.00 ± 2.33 | 6.87 ± 1.78 | 4.39 ± 0.66 | 2.20 ± 0.58 | 2.01 ± 0.60 | **8.06 ± 0.34** |
| Jumper-v0 | 2.71 ± 1.74 | -2.45 ± 0.30 | -2.49 ± 0.29 | 0.30 ± 0.04 | -0.04 ± 0.07 | 0.02 ± 0.08 | **3.15 ± 1.29** |
| Flipper-v0 | **35.55 ± 18.37** | 15.01 ± 13.22 | 13.00 ± 13.96 | 3.72 ± 0.53 | 0.71 ± 0.43 | 0.37 ± 0.16 | 26.96 ± 2.00 |
| Pusher-v0 | 7.40 ± 1.46 | 6.99 ± 1.30 | **11.16 ± 0.16** | 3.33 ± 0.73 | 0.94 ± 0.43 | 0.50 ± 0.29 | 7.77 ± 0.56 |
| Carrier-v0 | 4.33 ± 0.62 | -0.06 ± 0.04 | -0.14 ± 0.18 | 1.80 ± 0.25 | 0.11 ± 0.12 | 0.21 ± 0.13 | **5.84 ± 0.60** |
| UpStepper-v0 | 4.14 ± 0.39 | 3.18 ± 0.78 | 3.48 ± 0.91 | 1.78 ± 0.03 | 1.43 ± 0.04 | 1.38 ± 0.05 | **6.11 ± 2.11** |
| ObstacleTraverser-v0 | **6.02 ± 0.63** | 0.89 ± 0.41 | 0.62 ± 0.64 | 1.87 ± 0.28 | 1.48 ± 0.03 | 1.35 ± 0.10 | 2.90 ± 1.03 |
| Thrower-v0 | **3.01 ± 0.17** | 1.30 ± 0.39 | 2.96 ± 1.78 | 0.91 ± 0.01 | 0.61 ± 0.06 | 0.56 ± 0.14 | 1.66 ± 0.17 |
| PlatformJumper-v0 | **3.34 ± 1.97** | 1.51 ± 0.08 | 0.91 ± 0.78 | 1.66 ± 0.06 | 1.06 ± 0.34 | 1.36 ± 0.11 | 1.41 ± 0.26 |
| ObstacleTraverser-v1 | **4.92 ± 0.33** | 3.73 ± 1.08 | 2.94 ± 0.50 | 1.32 ± 0.04 | 1.06 ± 0.05 | 1.04 ± 0.10 | 3.64 ± 0.94 |
| Hurdler-v0 | **3.21 ± 0.45** | -0.02 ± 0.08 | 0.08 ± 0.27 | 0.71 ± 0.06 | 0.50 ± 0.01 | 0.46 ± 0.11 | 3.18 ± 1.90 |
| Traverser-v0 | **5.48 ± 0.80** | 3.10 ± 0.26 | 3.92 ± 0.96 | 2.77 ± 0.36 | 1.10 ± 0.60 | 1.14 ± 0.50 | 2.81 ± 0.16 |
| Carrier-v1 | 1.93 ± 0.33 | -0.15 ± 0.25 | -0.01 ± 0.01 | **2.50 ± 0.77** | 0.06 ± 0.27 | 0.03 ± 0.03 | 1.81 ± 0.47 |
| GapJumper-v0 | **4.05 ± 1.00** | 1.03 ± 2.79 | 2.58 ± 1.15 | 1.95 ± 0.05 | 1.36 ± 0.30 | 1.40 ± 0.12 | 2.34 ± 0.60 |

auto-encoder to embed rigid robots. We did not compare our method with GLSO because it requires some prior knowledge about design space that is not available in EvoGym and it cannot be directly used in multi-cellular robot design problems. In summary, our method has several advantages over GLSO. (1) Training cost. HERD uses a training-free embedding method while GLSO requires

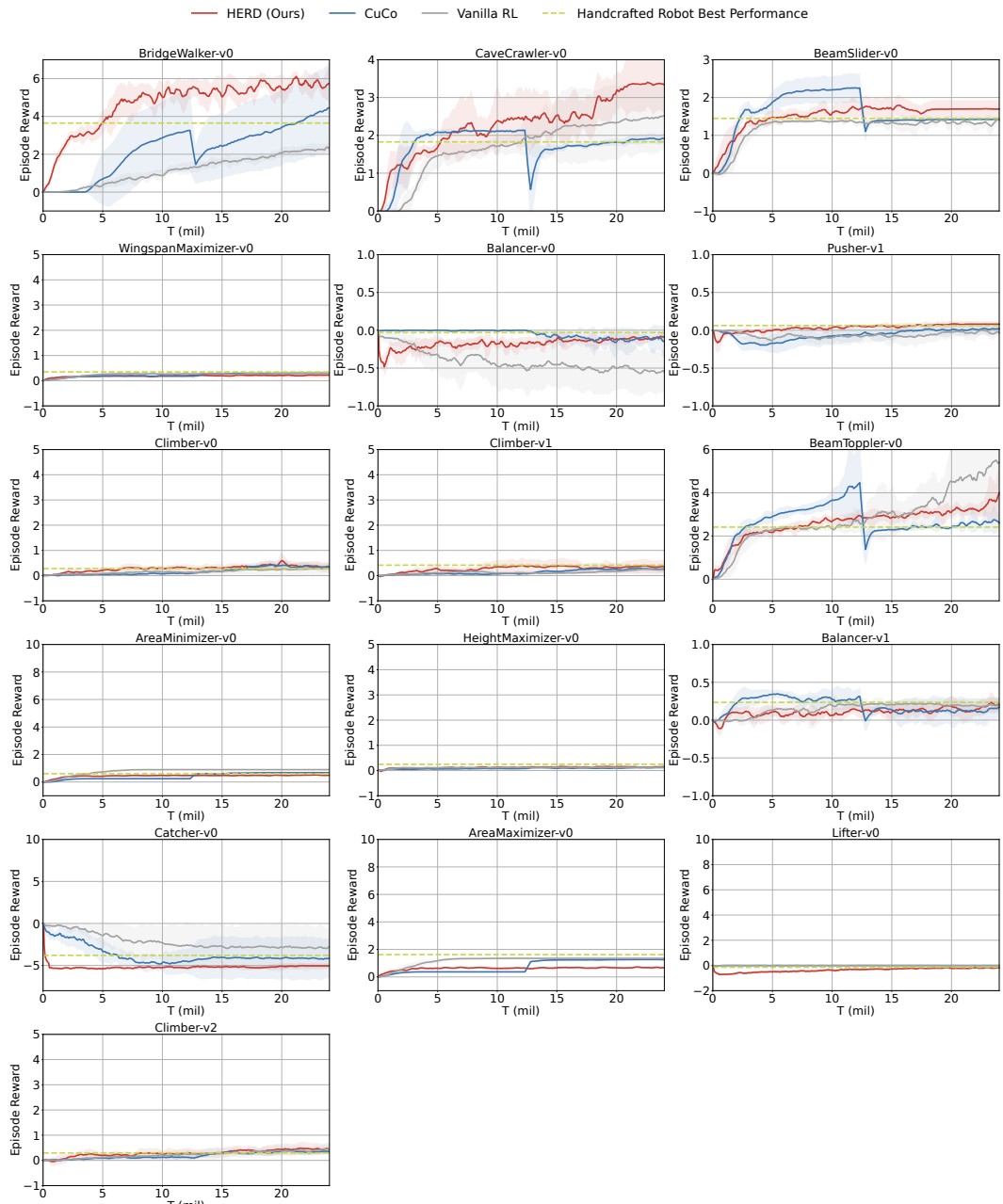

Figure 25: Performance of HERD in Other Tasks of EvoGym.

training a VAE, which could be computationally expensive and unstable. (2) Prior knowledge. HERD does not require as much domain-specific knowledge as GLSO does. Firstly, for data collection, GLSO adopts the set of grammar rules proposed in RoboGrammar Zhao et al. (2020), which requires prior knowledge of design space to produce a tractable and meaningful subspace of designs (Section 3.1 of the GLSO paper). Secondly, GLSO also co-train a proper prediction network (PPN) with VAE, and the training objective of PPN is also built on prior knowledge of the locomotion tasks they test (Section 3.4 of the GLSO paper). (3) Inductive bias. HERD explicitly encodes the hierarchical structure of design space, which could benefit the optimization process as we show in our experiment section, while GLSO does not consider any similar inductive bias.

Table 3: Quantitative results of our method HERD compared with various ablations.

| | HERD (Ours) | NGE | HERD w/o HE | HERD w/o C2F |
|---|---|---|---|---|
| Walker-v0 | **9.74 ± 0.81** | 7.16 ± 0.41 | 8.07 ± 0.32 | 2.76 ± 2.49 |
| DownStepper-v0 | **7.91 ± 0.76** | 2.49 ± 0.09 | 6.22 ± 0.61 | 3.21 ± 0.69 |
| Jumper-v0 | **2.71 ± 1.74** | 0.35 ± 0.20 | 0.03 ± 0.67 | -0.06 ± 0.22 |
| Flipper-v0 | **35.55 ± 18.37** | 28.08 ± 2.88 | 19.27 ± 8.98 | 1.06 ± 0.59 |
| Pusher-v0 | **7.40 ± 1.46** | 5.80 ± 1.83 | 5.45 ± 0.39 | 3.28 ± 1.52 |
| Carrier-v0 | 4.33 ± 0.62 | 3.52 ± 0.79 | **5.85 ± 1.88** | 1.81 ± 1.12 |
| UpStepper-v0 | **4.14 ± 0.39** | 1.32 ± 0.23 | 2.55 ± 1.31 | 2.49 ± 0.77 |
| ObstacleTraverser-v0 | **6.02 ± 0.63** | 1.64 ± 0.07 | 2.26 ± 0.26 | 1.45 ± 0.13 |
| Thrower-v0 | **3.01 ± 0.17** | 0.48 ± 0.01 | 0.80 ± 0.42 | 0.52 ± 0.15 |
| PlatformJumper-v0 | **3.34 ± 1.97** | 1.50 ± 0.14 | 1.50 ± 0.08 | 1.44 ± 0.11 |
| ObstacleTraverser-v1 | **4.92 ± 0.33** | 2.36 ± 0.27 | 2.00 ± 0.43 | 0.97 ± 0.16 |
| Hurdler-v0 | **2.69 ± 0.45** | 0.52 ± 0.06 | 0.88 ± 0.18 | 0.46 ± 0.08 |
| Traverser-v0 | **5.48 ± 0.80** | 2.28 ± 0.40 | 2.70 ± 0.10 | 2.57 ± 0.24 |
| Carrier-v1 | 1.93 ± 0.33 | 2.06 ± 0.87 | **2.43 ± 0.40** | 1.68 ± 0.16 |
| GapJumper-v0 | **4.05 ± 1.00** | 1.96 ± 0.10 | 1.95 ± 0.41 | 1.61 ± 0.38 |

