# OpenReview forum: "Leveraging Hyperbolic Embeddings for Coarse-to-Fine Robot Design"
_ICLR.cc/2024/Conference — ICLR 2024 poster_

### Official Review · Reviewer_Wtet · 2023-10-24

**Soundness:** 2 fair
**Presentation:** 3 good
**Contribution:** 2 fair
**Rating:** 6
**Confidence:** 4

**Summary:**

This paper investigates the soft cellular robot design problem. Instead of traditional approaches that apply genetic algorithms directly in the design space, this paper proposes to optimize the robot design in the robot embedding space, specifically a 2D hyperbolic space defined by Poincare ball. It constructs the embedding space by first sampling a set of robots in a coarse-to-fine procedure and applying Sarkar’s method to construct the Poincare ball. It then finds the optimal design from the sampled set by applying CEM in the hyperbolic space. During design optimization, it follows previous work [1] to gradually adapt and optimize a transformer-based policy via PPO to control the robot designs in the current CEM population.

**Strengths:**

1. Parameterizing the discrete robot design space into continuous embedding space and optimizing the robot in this continuous space is novel and interesting.

2. The experiments clearly show the advantages of the proposed NERD over baselines.

**Weaknesses:**

1. The generalizability of the proposed approach to other robot design representations is uncertain. Testing the approach on one common robot design representation would make it more convincing.

2. The choice of Sarkar’s construction for hyperbolic embedding needs more justification.

3. The experiments need improvements.

**Questions:**

1. The coarse-to-fine process seems specialized for the cellular robot representation in EvoGym. Is it generalizable to other robot design representations, such as articulate rigid robots (e.g. Wang et al. 2019 [1], Gupta 2021 [2])?

2. Since coarse-to-fine clustering is based on running K-Means with different random seeds, it seems to be possible that there are two robot designs located in different subtrees. Will such two designs be embedded in similar locations in the hyperbolic space?

3. Being relevant to the last question, it seems that the cell type is not used during the embedding. If I understand correctly, the embedding only captures the structural/subdivision similarity but not the cell functionalities. How can we interpret the distance in the embedding space into the similarity between designs?

4. Some notations are unnecessarily complicated. For example, it would be easier to understand if just saying the algorithm clusters the robot cells into 2, 4, 8, …, Nc/2, Nc components instead of using logarithm and exponential expressions.

5. Page 6 “In practice, similarity here means that fine-grained robots only need to change one component to be the same as their parent robots”. What does it mean? The fine-grained robots have more clusters than the parent robot, does it mean the fine-grained robots only change the cell type of one “new” cell cluster?

6. During the algorithm, is the robot design set fixed after the initial sampling? In other words, when you do $argmin$ in line 9 (Algorithm 1), do you always search for a design in the pre-sampled robot design pool?

7. After reading the paper, the phrase “coarse-to-fine” is indeed the way of building the candidate robot design pool instead of a coarse-to-fine optimization process. If this understanding is correct, the paper (especially the abstract and introduction) needs to be more clear about this point to prevent confusion.

8. EvoGym has 32 tasks however the algorithm is only tested on 15 of them. It would be great if the authors could evaluate the proposed algorithm on other tasks as well.

9. It would be great to provide the pseudo-code for the ablation variations to help understand the design of those ablation methods. For example, does HERD w/o C2F differ from full NERD mainly on the pre-sampled robot design set? So how does HERD w/o C2F sample the robot design set?
10. From the visual results, most of the optimized designs only have one single cell type. Does this result from the limitation of the embedding approach since the embedding only captures structural information but is not informative about the cell types?

11. How does the proposed embedding approach compare to other embedding approaches such as neural network embedding (e.g. GLSO [3])

[1] Tingwu Wang, Yuhao Zhou, Sanja Fidler, and Jimmy Ba. Neural graph evolution: Towards efficient automatic robot design

[2] Agrim Gupta, Linxi Fan, Surya Ganguli, and Li Fei-Fei. Metamorph: Learning universal controllers with transformers

[3] Jiaheng Hu, Julian Whitman, and Howie Choset. GLSO: Grammar-guided Latent Space Optimization for Sample-efficient Robot Design Automation

---

> ### Author Response · Authors · 2023-11-21
> **Additional Experimental Results, New Pseudo-Code for Ablation Variants and Clarifications to Other Questions [Part 1]**
>
> Thanks for the valuable comments and helpful suggestions. Here we provide additional experimental results and detailed explanations for your questions.
>
> > **Weakness 1 & Question 1**: The generalizability of the proposed approach to other robot design representations is uncertain. Testing the approach on one common robot design representation would make it more convincing. The coarse-to-fine process seems specialized for the cellular robot representation in EvoGym. Is it generalizable to other robot design representations, such as articulate rigid robots (e.g. Wang et al. 2019 [1], Gupta 2021 [2])?
>
> **A**: Though our paper mainly focuses on multi-cellular robot design problems, our method HERD can still be extended to other robot design representations, such as articulate rigid robots. Here we provide a solution to this setting.
>
> To implement the idea of coarse-to-fine robot design in rigid robots, the major challenge is to define the refinement process for coarse-grained robot designs. Once the refinement definition is given, we can organize the design space as a hierarchy through recursive refinement, embed the hierarchy in hyperbolic space, and use CEM for optimization as we did in the paper. In the following, we discuss the refinement of rigid robots in detail.
>
> Formally, a rigid robot design $D$ can be represented by a tree $D=(V,E,g,h)$, where each vertex $v\in V$ represents a joint in robot morphology and each edge $e=(v_i,v_j)\in E$ represents a limb connecting two joints $v_i,v_j$ and $|E|=|V|-1$. The function value $g(v)$ defines joint-specific attributes and function value $h(v_i,v_j)$ defines limb-specific attributes.
>
> The intuition behind the solution is to regard the coarse-grained robot as a simplified tree, to which we enrich details such as adding new vertices and new edges during refinement, such that the fine-grained robot has a 'similar' shape to the coarse-grained robot. Given a design $D$, we denote the finer-grained design of $D$ by $D'=(V',E',g',h')$. $D'$ is a valid refinement of $D$ if and only there is a vertex map $\pi:V'\to V$ such that the following conditions are satisfied:
>
> (i) (coarse-to-fine condition): $|V|<|V'|$;
>
> (ii) (edge condition): the edge $e'=(v'_i,v'_j)\in E'$ if and only if there is an edge $e=(v_1,v_2)\in E$ such that $\pi(v'_i)=v_i$ and $\pi(v'_j)=v_j$;
>
> (iii) (vertex attribute condition): for each $v'\in V'$, we have $g'(v')=g(\pi(v'))$;
>
> (iv) (edge attribute condition): for each $e'=(v'_i,v'_j)\in E'$, we have $h'(v_i',v_j')=h(\pi(v_i'),\pi(v_j'))$;
>
> The above conditions enable $D'$ to be one of the refinements of $D$ with a 'similar' shape. Given this refinement definition, we can then organize the design space of rigid robots as a hierarchy like multi-cellular robots and use the hyperbolic solution for coarse-to-fine optimization.
>
> Due to the time limitation of rebuttal, we only provide a conceptual solution. We believe this could be a promising future work.
>
> > **Weakness 2**: The choice of Sarkar's construction for hyperbolic embedding needs more justification.
>
> **A**: There are two reasons for this choice. Firstly, Sarkar's construction can embed the tree structure with arbitrarily low distortion [5], which matches our needs of embedding the tree of robot designs. Secondly, Sarkar's construction is a training-free method that will not bring extra computational cost and is also simple to implement.
>
>
> > **Weakness 3**: The experiments need improvements.
>
> **A**: Thank you for your suggestions. We have improved our experiments by including the results of other tasks from EvoGym in Figure 25 of Appendix C in the revised reversion of our paper. Please refer to Question 8 for extra discussions.
>
>
> > **Question 2**: Since coarse-to-fine clustering is based on running K-Means with different random seeds, it seems to be possible that there are two robot designs located in different subtrees. Will such two designs be embedded in similar locations in the hyperbolic space?
>
> **A**: There might be some misunderstandings. For each number of clusters, we only run K-Means *once* on robot cells per experiment. In this case, two similar robot designs will always be located in the same subtrees, then they will also be embedded in similar locations in the hyperbolic space.

---

> > ### Author Response · Authors · 2023-11-21
> > **Additional Experimental Results, New Pseudo-Code for Ablation Variants and Clarifications to Other Questions [Part 2]**
> >
> > > **Question 3**: Being relevant to the last question, it seems that the cell type is not used during the embedding. If I understand correctly, the embedding only captures the structural/subdivision similarity but not the cell functionalities. How can we interpret the distance in the embedding space into the similarity between designs?
> >
> > **A**: There might be some misunderstandings. The similarity considering cell types is captured in the robot hierarchy and the cell functionalities will also be encoded in the embeddings. Specifically, to capture their similarity, during the construction of the robot hierarchy, the child robot only changes one component compared to its parent, which leads to a high similarity of the child robots originating from the same parent robot. More generally, the similarity of any two robots in the robot hierarchy is largely determined by their distance to the nearest common ancestor robot node. If they are close to their nearest common ancestor robot node, e.g., the nearest common ancestor robot node is their shared parent robot node, they tend to be similar. This similarity is also encoded in our hyperbolic embeddings thanks to the ability of Sarkar's Construction to embed hierarchy with low distortion error.
> >
> >
> > > **Question 4**: Some notations are unnecessarily complicated. For example, it would be easier to understand if just saying the algorithm clusters the robot cells into 2, 4, 8, …, Nc/2, Nc components instead of using logarithm and exponential expressions.
> >
> > **A**: Thanks for the helpful suggestion, we have fixed this problem in the revised version of our paper.
> >
> >
> > > **Question 5**: Page 6 “In practice, similarity here means that fine-grained robots only need to change one component to be the same as their parent robots”. What does it mean? The fine-grained robots have more clusters than the parent robot, does it mean the fine-grained robots only change the cell type of one “new” cell cluster?
> >
> > **A**: Yes, we only change the cell type of clusters of fine-grained robots. This is necessary to make child fine-grained robots similar to their parent coarse-grained robots.
> >
> >
> > > **Question 6**: During the algorithm, is the robot design set fixed after the initial sampling? In other words, when you do $\arg\min$ in line 9 (Algorithm 1), do you always search for a design in the pre-sampled robot design pool?
> >
> > **A**: Yes, we do $\arg\min$ in the pre-sampled robot design pool that is embedded in the Poincaré ball before robot design optimization.
> >
> > > **Question 7**: After reading the paper, the phrase “coarse-to-fine” is indeed the way of building the candidate robot design pool instead of a coarse-to-fine optimization process. If this understanding is correct, the paper (especially the abstract and introduction) needs to be more clear about this point to prevent confusion.
> >
> > **A**: There might be some misunderstandings. Our optimization is also a coarse-to-fine process but in an automatic way. Though we do not add any other mechanisms to force the robots to be finer-grained, our method gradually increases the granularity of robots solely for better performance. This phenomenon is shown in Figure 5 of our paper: as the optimization proceeds, our method samples robots from the center to the border in Poincaré ball, which is exactly the process of coarse-to-fine robot design.
> >
> >
> > > **Question 8**: EvoGym has 32 tasks however the algorithm is only tested on 15 of them. It would be great if the authors could evaluate the proposed algorithm on other tasks as well.
> >
> > **A**: Thanks for the constructive suggestion. We have included the results of other tasks in Figure 25 of Appendix C in the revised version of our paper. Considering all the tasks in EvoGym, HERD outperforms all the other robot design baselines (CuCo, Vanilla RL, BO, GA, CPPN-NEAT) in 15 tasks and achieves a comparable performance in 10 tasks. To our best knowledge, our work is the first one to solve this many tasks in EvoGym.
> >
> >
> > > **Question 9**: It would be great to provide the pseudo-code for the ablation variations to help understand the design of those ablation methods. For example, does HERD w/o C2F differ from full HERD mainly on the pre-sampled robot design set? So how does HERD w/o C2F sample the robot design set?
> >
> > **A**: Thanks for the constructive suggestion. We have provided the pseudo-code of the ablation variants in Algorithm 3 and 4 in Appendix B.4. Please refer to the revised version of our paper for details.
> >
> > HERD w/o C2F differs from full HERD mainly in not using the idea of coarse-to-fine. It is true that HERD w/o C2F does not use the pre-sampled robot set, instead, it maintains a distribution function that directly represents the probability of each robot cell's type. Then it uses CEM to optimize this distribution.

---

> > > ### Author Response · Authors · 2023-11-21
> > > **Additional Experimental Results, New Pseudo-Code for Ablation Variants and Clarifications to Other Questions [Part 3]**
> > >
> > > > **Question 10**: From the visual results, most of the optimized designs only have one single cell type. Does this result from the limitation of the embedding approach since the embedding only captures structural information but is not informative about the cell types?
> > >
> > > **A**: Not really. As we discussed in Question 3, our embedding approach also captures the information about the cell types.
> > >
> > > This phenomenon is a proper result of optimization. As we show in Figure 5, our method has explored many robots including these single-type robots, which turned out to be the most suitable robot for the given tasks. Also please note that only coarser-grained robots tend to have one single cell type such as task PlatformJumper-v0 in Figure 3, while finer-grained robots tend to have multiple cell types such as task Carrier-v1 in Figure 3.
> > >
> > >
> > > > **Question 11**: How does the proposed embedding approach compare to other embedding approaches such as neural network embedding (e.g. GLSO [3])
> > >
> > > **A**: Thanks for pointing out this paper. Our method HERD has several advantages over GLSO.
> > >
> > > (1) Training cost. HERD uses a training-free embedding method while GLSO requires training a VAE, which could be computationally expensive and unstable.
> > > (2) Prior knowledge. HERD does not require as much domain-specific knowledge as GLSO does. Firstly, for data collection, GLSO adopts the set of grammar rules proposed in RoboGrammar [4], which requires prior knowledge of design space to produce a tractable and meaningful subspace of designs (Section 3.1 of the GLSO paper). Secondly, GLSO also co-train a proper prediction network (PPN) with VAE, and the training objective of PPN is also built on prior knowledge of the locomotion tasks they test (Section 3.4 of the GLSO paper).
> > > (3) Inductive bias. HERD explicitly encodes the hierarchical structure of design space, which could benefit the optimization process as we show in our experiment section, while GLSO does not consider any similar inductive bias.
> > >
> > > We have added the above discussion in Appendix E of the revised version of our paper.
> > >
> > > Thanks again for your efforts and insightful comments! We hope our clarification addresses your concerns and sincerely appreciate it if you could re-evaluate our work. Any further feedback and discussions are much appreciated.
> > >
> > > ---
> > >
> > > **References**
> > >
> > > [1] Tingwu Wang, Yuhao Zhou, Sanja Fidler, and Jimmy Ba. Neural graph evolution: Towards efficient automatic robot design
> > >
> > > [2] Agrim Gupta, Linxi Fan, Surya Ganguli, and Li Fei-Fei. Metamorph: Learning universal controllers with transformers
> > >
> > > [3] Jiaheng Hu, Julian Whitman, and Howie Choset. GLSO: Grammar-guided Latent Space Optimization for Sample-efficient Robot Design Automation
> > >
> > > [4] Zhao, Allan, Jie Xu, Mina Konaković-Luković, Josephine Hughes, Andrew Spielberg, Daniela Rus, and Wojciech Matusik. "Robogrammar: graph grammar for terrain-optimized robot design." ACM Transactions on Graphics (TOG) 39, no. 6 (2020): 1-16.
> > >
> > > [5] Frederic Sala, Chris De Sa, Albert Gu, and Christopher R  ́e. Representation tradeoffs for hyperbolic embeddings. In International conference on machine learning, pp. 4460–4469. PMLR, 2018

---

> ### Comment · Reviewer_Wtet · 2023-11-22
> **Reply to Author's Response**
>
> Thank you for your detailed response. The additional results from the whole EvoGym tasks are helpful and thank you for the effort to put them together. After reading the response, I have two follow-up questions:
>
> 1. The answer to Question 2 confuses me further. Does running K-Means only once per experiment mean you only have one robot for each number of clusters? In other words, each robot design only has one child fine-grained design, is that the case? Otherwise, how do you get different child designs for one parent design with only one K-means run? More details about how this step works would be super helpful and is also relevant to my Question 3.
>
> 2.  For the reply to my Question 7, I do think I understand correctly. The coarse-to-fine is mainly for constructing the design candidate pool instead of optimization. The optimization process shown in Algorithm 1 can be "automatic coarse-to-fine" but there is no theoretical guarantee. It basically finds the nearest design in the embedding space around the current CEM mean and it can be possible to get coarse-grained robots in the later stage of the algorithm. Feel free to point out if I misunderstood anything here or if there is any other missing special mechanism designed to enforce the coarse to fine optimization.

---

> > ### Author Response · Authors · 2023-11-22
> > **Thank you for the feedback**
> >
> > Thanks for the timely reply. Here we provide detailed explanations to your questions.
> >
> > > **Question 1**: The answer to Question 2 confuses me further. Does running K-Means only once per experiment mean you only have one robot for each number of clusters? In other words, each robot design only has one child fine-grained design, is that the case? Otherwise, how do you get different child designs for one parent design with only one K-means run? More details about how this step works would be super helpful and is also relevant to my Question 3.
> >
> > **A**: We would like to clarify that (1) we have multiple robots for each number of clusters and (2) each parent robot design has multiple fine-grained child designs as well, which will be explained in detail below. First, recall that we run K-Means on robot cells to define the granularity of robot designs, i.e., the subspace of the robot designs, which usually has a low degree of freedom and is much smaller than the original design space. Coarse-grained robots have a smaller number of clusters than fine-grained robots. A simplified example is shown in Figure 2(a), where coarse-grained robots $D_1, D_2$ only have one cluster of robot cells, and fine-grained robots $\mathcal{C}(D_1),\mathcal{C}(D_2)$ have two clusters of robot cells.
> >
> > (1) We have multiple robots for each number of clusters, which corresponds to a specific granularity of robot designs. This is because though which robot cells are clustered is determined after running K-Means, the cell type for each cluster of robot cells is undecided. Since we have five cell types in EvoGym, there will be up to $5^K$ different robot designs if the number of clusters is $K$. Figure 2(a) Box A shows multiple robot designs under the same clustering result, i.e., only two clusters: {left-cluster, right-cluter}.
> >
> > (2) Each parent robot design has multiple fine-grained child designs. This is because though we run K-Means only once for each granularity (i.e., each number of clusters) per experiment, the clustering results are reused *multiple* times during the construction of the robot hierarchy. Specifically, to get different child designs for one parent design, we perform the following operations *multiple* times: initialize a child design by inheriting the parent design, divide the child design into several clusters of robot cells according to the clustering result for finer-grained robots, and change the cell type of a randomly selected cluster of the child design. Figure 2(a) can serve as an intuitive example. Given the clustering results for child robots {left-cluster, right-cluter}, the coarse-grained design $D_1$ can have multiple children fine-grained designs as shown in Box A below design $D_1$.
> >
> >
> > > **Question 2**: For the reply to my Question 7, I do think I understand correctly. The coarse-to-fine is mainly for constructing the design candidate pool instead of optimization. The optimization process shown in Algorithm 1 can be "automatic coarse-to-fine" but there is no theoretical guarantee. It basically finds the nearest design in the embedding space around the current CEM mean and it can be possible to get coarse-grained robots in the later stage of the algorithm. Feel free to point out if I misunderstood anything here or if there is any other missing special mechanism designed to enforce the coarse to fine optimization.
> >
> > **A**: There might be different understandings regarding the definition of the coarse-to-fine optimization process. The first understanding, as suggested by the reviewer, is to explicitly enforce the coarse-to-fine optimization, which is not the case in our work. The second understanding, as we do in our paper, is that it is indeed a coarse-to-fine robot design method as long as the optimization trajectory shows an increased granularity together with improved performance, though the granularity direction is not explicitly specified. It is also worth noting that the first understanding to explicitly enforce the optimization to be coarse-to-fine may harm performance because (1) the coarse-to-fine interval is hard to control and (2) not all tasks require the finest-grained robots.
> >
> > Thanks for the suggestion and to avoid ambiguity, we will add some discussions regarding the definition of the coarse-to-fine optimization process in the revised version of our paper.
> >
> >
> > Thanks again for your timely reply! We hope our new clarification addresses your questions and any further feedback and discussions are much appreciated.

---

> ### Comment · Reviewer_Wtet · 2023-11-22
> **Reply to Authors' Response**
>
> Thank you for your prompt response and further clarification on the robot construction process. Now I think it is clear to me how this process works. If I understand correctly, the short answer is that the cluster layout of all child robots under a parent robot are all the same while the only difference among them is their different choices of cell types. Also please update your manuscript about this detail, since I cannot find any explicit paragraph about this detail after I read the method section several times). Given this fact, I have one last question left.
>
> Since you only run K-means once for each parent robot design to construct finer robot layouts, will it miss some large branches of robot structures and can it be a limitation of the proposed method? For example, assume we have 4x4 grid cells, if our first K-means cluster the cell into left and right clusters such as (A, B denote for different cell types)
>      $$AABB$$
>      $$AABB$$
>      $$AABB$$
>      $$AABB$$
>     Is it still possible for the algorithm to get the robot in a top-down layout in its descendants as
>      $$AAAA$$
>      $$AAAA$$
>      $$BBBB$$
>      $$BBBB$$
>     Since each round of the algorithm is only able to change the cell type of one cluster, it seems this structure branch will be missed and can limit the optimality of the algorithm.

---

> ### Author Response · Authors · 2023-11-23
> **Thank you for the extra feedback**
>
> Thank you for your prompt reply! Here we provide additional explanations to your questions.
>
> > **Suggestion 1**: If I understand correctly, the short answer is that the cluster layout of all child robots under a parent robot is all the same while the only difference among them is their different choices of cell types. Also please update your manuscript about this detail, since I cannot find any explicit paragraph about this detail after I read the method section several times).
>
> **A**: This is a correct understanding. Thanks for the suggestion, we have updated our paper about this detail in Section 4.1.
>
> > **Question 1**: Since you only run K-means once for each parent robot design to construct finer robot layouts, will it miss some large branches of robot structures and can it be a limitation of the proposed method? For example, assume we have 4x4 grid cells, if our first K-means cluster the cell into left and right clusters such as (A, B denote for different cell types)
> > $$
> > \begin{matrix} AABB \\\ AABB \\\ AABB \\\ AABB \end{matrix}
> > $$
> > Is it still possible for the algorithm to get the robot in a top-down layout in its descendants as
> > $$
> > \begin{matrix} AAAA\\\ AAAA\\\ BBBB\\\ BBBB \end{matrix}
> > $$
> > Since each round of the algorithm is only able to change the cell type of one cluster, it seems this structure branch will be missed and can limit the optimality of the algorithm.
>
> **A**: We would like to thank the reviewer for pointing this out and here we provide some analysis into this issue. In short, part of this problem may occur, but it will not affect the optimization too much and can be easily solved.
>
> During our implementation, we did observe this issue. However, it is caused by the small number of mutations (the number of clusters to change), which is a hyperparameter. Actually, if we increase the value of this hyperparameter, we will be able to cover the entire design space definitely. For the sake of maintaining the similarity between parent robots and child robots, we set the value to $1$. There is a trade-off between this similarity and the coverage of design space. In the empirical study, we found that the final performance was sufficiently good when the number of mutations was set to $1$.
>
> Apart from that, even though this hyperparameter is set to $1$, the robot hierarchy might still include most of Robot-1's child robots shown as follows:
> $$
> \text{(Robot-1):}
> \begin{matrix} AAAA\\\ AAAA\\\ BBBB\\\ BBBB \end{matrix}\quad
> \text{(Children):}
> \begin{matrix} BBAA\\\ BBAA\\\ BBBB\\\ BBBB \end{matrix}\quad
> \begin{matrix} AABB\\\ AABB\\\ BBBB\\\ BBBB \end{matrix}\quad
> \begin{matrix} AAAA\\\ AAAA\\\ AABB\\\ AABB \end{matrix}\quad
> \begin{matrix} AAAA\\\ AAAA\\\ BBAA\\\ BBAA \end{matrix}
> $$
> These child robots can be generated by mutating $2\times 2$ cells of the following left-right robots that are included in the current robot hierarchy:
> $$
> \begin{matrix} AABB\\\ AABB\\\ AABB\\\ AABB \end{matrix}\quad
> \begin{matrix} BBAA\\\ BBAA\\\ BBAA\\\ BBAA \end{matrix}
> $$
>
> Thanks again for your efforts and insightful comments! We hope our clarification addresses your concerns and sincerely appreciate it if you could re-evaluate our work.

---

### Official Review · Reviewer_S1iZ · 2023-11-01

**Soundness:** 3 good
**Presentation:** 4 excellent
**Contribution:** 3 good
**Rating:** 8
**Confidence:** 3

**Summary:**

The paper proposes a novel co-design algorithm for multi-cellular robots. The authors noticed two critical insights 1. "coarse-grained robots are less complicated to design and control due to their low degree of freedom" and 2. "coarse-grained robots can usually solve part of the task". Built on these two insights, the authors proposed a coarse-to-fine design algorithm where a CEM optimizer optimize robot design in Poincare ball hyperbolic embedding space. The hyperbolic embedding embeds a coarse-to-fine tree structure so the optimization effectively. The authors benchmark the proposed algorithm on a variety of environments in evo-gym and shows the effectiveness of the algorithm.

**Strengths:**

- The writing quality of this paper is very good. All design choices are introduced with clear motivation. Ideas are explained clearly with text and high-quality figures.
- It's quite novel for the authors to map the robot design to a space that has a favorable optimization landscape for methods as simple as CEM.
- The two insights the authors point out, 1. "coarse-grained robots are less complicated to design and control due to their low degree of freedom" and 2. "coarse-grained robots can usually solve part of the task", has a potential to motivate new research
- The proposed method is a sound algorithm  to the cellular robot design problem and the experiments thoroughly justified the point.

Overall this is a technically solid paper with novel contributions that's clearly above acceptance threshold.

**Weaknesses:**

- It's unclear how general is the hyperbolic space is for robotics design. See my questions in the next section
- It's unclear to me to me how exactly any point on the manifold correspond to a robot configuration. Some concrete example may help understanding. If I randomly sample a point using a coordinate, what's the robot configuration it correspond to? It seems that human has to handcraft such rules.

**Questions:**

- The authors chose Poincare ball as the hyperbolic embedding space for the design tree. I am wondering how general is this? In particular, hyperbolic manifold seems very limiting. For example, let's say the robot can be comprised of material A and material B. From the center, I extend two different trees branching towards material A and material B separately. However, in each branch, fine-grained branches start to contain configurations that blends A & B which means the two branches should merge. In this case, you will have to have a finite cylinder manifold, correct?

- How exactly does the manifold correspond to designs or robot parameters? How many branches are there? I understand how do you map between the euclidean space and the hyperbolic space but this seems unclear. Giving more visualizations across the disk manifold will be helpful.

---

> ### Author Response · Authors · 2023-11-21
> **New Visualization Results and Clarifications of Other Questions**
>
> Thanks for your positive comments and constructive suggestions. Here we provide extra visualization results and detailed explanations for your questions.
>
> > **Weakness 1**: It's unclear to me how exactly any point on the manifold corresponds to a robot configuration. Some concrete examples may help to understand. If I randomly sample a point using a coordinate, what's the robot configuration it corresponds to? It seems that human has to handcraft such rules.
>
> **A**: We would like to clarify that we do not have to handcraft such rules. Instead, we simply calculate the distance between the sampled point and other points that are embedded in the hyperbolic manifold. Then we choose the point and its corresponding robot design with minimum distance. Please refer to Figure 21 in Appendix B.2 in the revised version of our paper for a simplified visualization.
>
>
> > **Question 1**:  It's unclear how general the hyperbolic space is for robotics design. The authors chose the Poincare ball as the hyperbolic embedding space for the design tree. I am wondering how general this is. In particular, the hyperbolic manifold seems very limiting. For example, let's say the robot can be comprised of material A and material B. From the center, I extend two different trees branching towards material A and material B separately. However, in each branch, fine-grained branches start to contain configurations that blend A & B which means the two branches should merge. In this case, you will have to have a finite cylinder manifold, correct?
>
> **A**: Thanks for the idea. In our experiment, the "branch-merge problem" might happen with a low possibility but will not influence the optimization severely. This is because our method HERD optimizes the robot designs starting from the center of the Poincaré ball, and it can automatically identify the promising sub-space regardless of whether the branches are merged or not. The idea of the finite cylinder manifold might further improve final performance, but optimizing within a three-dimensional space may also introduce additional complexities to the problem, which requires in-depth empirical investigation.
>
>
> > **Question 2**: How exactly does the manifold correspond to designs or robot parameters? How many branches are there? I understand how you map between the Euclidean space and the hyperbolic space but this seems unclear. Giving more visualizations across the disk manifold will be helpful.
>
> **A**: To correspond a point in the manifold to a design, as we discussed in Weakness 1, we simply choose the pre-embedded robot design in the manifold with the minimum distance to the point.
>
> The number of branches may vary with the level of the hierarchy of robot designs. For example, the number of branches originating from the root node only has two branches. As for the other level of the hierarchy, we restrict the number of branches originating from the same non-leaf node to $20$. This restriction may limit the search space, but since the hierarchy grows exponentially, the search space is still large enough to contain the designs that solve given tasks, as long as the number of branches exceeds a threshold.
>
> Thanks for the suggestion of visualization. We have visualized the 'corresponding process' in a simplified disk manifold, as there are too many points in the original manifold to be visualized clearly. Please refer to Figure 21 in Appendix B.2 in the revised version of our paper for details

---

> ### Comment · Reviewer_S1iZ · 2023-11-23
>
> Thank you for your clarifications and additional visualizations.
>
> I acknowledge that I've read your responses carefully as well peer reviewers' comments. After reading through reviewer Wtet's comments carefully, I realized there were a few components that I didn't fully understand in my first pass. However, your responses are very helpful to give me a better under standing.
>
> Overall I think the merits of the paper still outweight its problems. On the other hand due to the reason below I decide to lower my confidence to 3.

---

### Official Review · Reviewer_9c5Z · 2023-11-01

**Soundness:** 3 good
**Presentation:** 4 excellent
**Contribution:** 3 good
**Rating:** 6
**Confidence:** 3

**Summary:**

This paper proposes a coarse-to-fine method for designing multi-cellular robots with hyperbolic geometry.

**Strengths:**

1. The paper is well written, easy to follow, the visualizations are nicely designed and the accompanying website shows impressive results.
2. Incorporating hyperbolic geometry into robot planning is innovative and interesting.
2. The experiment shows promising results over a variety of tasks.

**Weaknesses:**

1. As one of the concerns of multi-cellar system, the design space is very large, so how hyperbolic solution solved this problem is not well examined.
2. The quantitative analysis of the proposed model is somewhat absent, some tables will help the understanding of the evaluation of different methods.

**Questions:**

please see the weaknesses

---

> ### Author Response · Authors · 2023-11-21
> **New Quantitative Analysis and Clarifications of Other Questions**
>
> Thanks for your comments and valuable suggestions. Here provide additional quantitative analysis of the proposed method and explanations for your questions.
>
> > **Weakness 1**: As one of the concerns of the multi-cellar system, the design space is very large, so how a hyperbolic solution solves this problem is not well examined.
>
> **A**: Our method HERD solves the large space problem of multi-cellular system by adopting a coarse-to-fine robot design mechanism, in which HERD starts by finding the optimal coarse-grained robot designs and subsequently refines them. For a simple and better optimization, we embed any-grained robots in hyperbolic space and use Cross-Entropy Method to optimize robot designs in hyperbolic space. The design space is substantially reduced in this way because (1) the design space of coarse-grained robots that HERD optimizes first is much smaller than the original design space; (2) by using the guidance of coarse-grained designs, HERD focuses on promising sub-space without the needs to explore the whole space, which further reduces the design space. This intuition is elaborated in Figure 1 and the Introduction Section of our paper.
>
> We have also examined the above solution empirically in our paper. We first show that HERD can successfully solve plenty of tasks in Figures 3 & 22, then we check the functionality of coarse-to-fine and hyperbolic embedding components in Figure 4, finally, we visualize the optimization process of HERD in Figure 5 for an in-depth understanding of how it works.
>
>
> > **Weakness 2**: The quantitative analysis of the proposed model is somewhat absent, some tables will help the understanding of the evaluation of different methods.
>
> **A**: Thanks for the suggestion. We have revised our paper and updated several tables of the performance comparisons with baselines and ablation studies in Appendix D. Specifically, we compare our method HERD with various baselines in Table 2 and HERD outperforms other robot design methods in most tasks, which showcases the effectiveness of our method. We also display the numeric results of ablation studies in Table 3 in Appendix D, which demonstrates the validity of the main components of our method: coarse-to-fine robot design and hyperbolic embedding. Please refer to the updated version of our paper for detailed quantitative results.

---

> > ### Comment · Reviewer_9c5Z · 2023-11-22
> >
> > Thank you for the explanation.
> >
> > I understand conceptually that a coarse-to-fine design reduces computational complexity. All evaluations presented are qualitative, as indicated by the authors in Figures 3, 4, 5, and 22. Demonstrating quantitatively how much computation was saved would further enhance the significance of this work.
> >
> > I will stay (slightly) positive to this paper.

---

### Official Review · Reviewer_WvQM · 2023-11-02

**Soundness:** 2 fair
**Presentation:** 3 good
**Contribution:** 2 fair
**Rating:** 6
**Confidence:** 3

**Summary:**

This paper addresses the task of multi-cellular soft robot design given a specific robotic task in simulation.

It proposes a novel robot design optimization method to enable efficient optimal design search in a vast robot design space. The proposed method searches the design space in a coarse-to-fine manner. The design space is first transformed into a hyperbolic space, where each robot design is embedded into the hyperbolic space via a train-free strategy. The finer design has a larger embedding norm, forming a coarse-to-fine robot hierarchy in the hyperbolic space. Then, the Cross-Entropy Method is adopted to search in the hyperbolic space.

This paper uses 15 tasks from EvoGym for experiments. It compares with manual design and several robot design methods. It also conducts ablations on the coarse-to-fine and hyperbolic embeddings.

**Strengths:**

1. This paper is well-written.
2. The experiment section provides enough evaluations, i.e., 15 diverse tasks in the simulation.
2. Searching robot design space in a coarse-to-fine manner via hyperbolic space is novel.

**Weaknesses:**

1. The issue of local optimal (not very good performance compared to other baselines) and large variance: Does the parent node of an optimal robot design node consistently outperform the parent node of a non-optimal robot design node? If not, coarse-to-fine seems greedy and vulnerable to local optimal. The proposal method, HERD, appears to have no mechanism for exploration in the global design space, potentially leading to the high variance in Figure 3. And HERD is not significantly better than baselines.

**Questions:**

1. What are the policy architecture and learning methods adopted for the baselines? Are they the same as those of HERD?
2. HEARD w/o HE:  how to integrate NGE with CEM? Could you provide more details on this ablation?
3. How is the performance of a HandCrafted robot determined？ Does it use a transformer-PPO policy? If yes, it should also be reported with variance, e.g., in Figure 3.
4. How can we guarantee that sequential refinements on a coarse design are better than sequential refinements on another coarse design, given that the former coarse design outperforms the latter coarse design?

---

> ### Author Response · Authors · 2023-11-21
> **New Visualization Results, New Pseudo-Code for Ablation Variants, and Clarifications of the Questions [Part 1]**
>
> Thanks for the comments and helpful suggestions. Here we provide detailed clarifications to your questions.
>
> > **Weakness 1**: The issue of local optimal (not very good performance compared to other baselines). HERD is not significantly better than baselines.
>
> **A**: We respectfully disagree with this claim. Our method HERD is significantly better than baselines as shown in Figure 3 and Figure 22 of the paper, which includes 15 tasks. In summary, (1) HERD outperforms all the other robot design baselines (CuCo, Vanilla RL, BO, GA, CPPN-NEAT) in 12 tasks out of 15 tasks; (2) HERD is much better than HandCrafted Robot in 7 tasks and has a comparable performance in 5 tasks.
>
> HERD utilizes a variant of the Cross-Entropy Method (CEM) for optimizing the coarse-to-fine robot design process. CEM is a popular stochastic optimization method due to its simplicity and effectiveness. Because of its population-based search and stochastic nature, it has less tendency to local optimality convergence than many other heuristic methods. Its global optimality can be further enhanced by using a mixture of Gaussian models [1], which can be an interesting direction for future work.
>
>
> > **Weakness 2**: The proposal method, HERD, appears to have no mechanism for exploration in the global design space.
>
> **A**: There might be some misunderstandings. HERD has an exploration mechanism, which is built inherently in the Cross-Entropy Method (CEM) -- a stochastic optimization method we use. Concretely, HERD maintains a distribution of robot designs and optimizes the distribution with CEM. When sampling from this distribution, HERD explores the robot design space such that robots of lower performance will still be sampled, though with a relatively lower possibility.
>
>
> > **Weakness 3**: The issue of high variance in Figure 3.
>
> **A**: We appreciate the reviewer pointing this out. High variance is a common challenge in the field of robot design. For example, GA in task Carrier-v1 and CuCo in task GapJumper-v0 also show high variance. One major cause of this problem is that the optimal control policies are possibly difficult to learn as the robot designs are changing over time. Similar robots might have totally different performances due to the learning issue of control policies. For a fair comparison, our method HERD together with baselines CuCo, Vanilla RL, and HandCrafted Robot adopt the same control policy architecture (transformers) and optimization algorithm (PPO). It is an open and important direction for future work to investigate more effective control policy learning methods in the field of robot design.
>
> > **Weakness 4 & Question 4**: Does the parent node of an optimal robot design node consistently outperform the parent node of a non-optimal robot design node? If not, coarse-to-fine seems greedy and vulnerable to local optimal. How can we guarantee that sequential refinements on a coarse design are better than sequential refinements on another coarse design, given that the former coarse design outperforms the latter coarse design?
>
> **A**: We do not need to guarantee that "the parent node of an optimal robot design node consistently outperforms the parent node of a non-optimal robot design node", because our method HERD is not a greedy deterministic algorithm. As we discussed in Weakness 2, HERD uses a stochastic optimization method to optimize the distribution of robot design instead of one or several robot samples. Hence, HERD can still explore child nodes from another subtree if the current child nodes are less efficient.
>
>
> > **Question 1**: What are the policy architecture and learning methods adopted for the baselines? Are they the same as those of HERD?
>
> **A**: The baselines CuCo, Vanilla RL, and HandCrafted Robot adopt the same control policy architecture (transformers) and learning method (PPO) as our method HERD. For the other baselines GA, BO, and CPPN-NEAT, we use their official implementations from CuCo and EvoGym.

---

> > ### Author Response · Authors · 2023-11-21
> > **New Visualization Results, New Pseudo-Code for Ablation Variants, and Clarifications of the Questions [Part 2]**
> >
> > > **Question 2**: HERD w/o HE: how to integrate NGE with CEM? Could you provide more details on this ablation?
> >
> > **A**: We would like to clarify that HERD w/o HE does not integrate NGE with CEM, but combines NGE with the idea of coarse-to-fine robot design. This variant removes hyperbolic embeddings and CEM of our method HERD and uses NGE to implement coarse-to-fine robot design. We use this variant to show the effectiveness of hyperbolic embeddings. Concretely, we re-implement the original NGE in the EvoGym benchmark based on their released code. The core modification is the *mutation procedure*. To enable multi-cellular robot design, we randomly change the type of each robot cell with probability $0.1$, where the types are chosen from {Empty, Rigid, Soft, Horizontal Actuator, Vertical Actuator}. To enable coarse-to-fine robot design, we refine each robot with a probability of $0.2$ if it is coarse-grained. Then the mutated robots are viewed as offsprings and added to the robot population of NGE.
> >
> > Another variant HERD w/o C2F does not adopt the idea of coarse-to-fine but uses CEM to directly search the optimal design. This ablation can verify the effectiveness of coarse-to-fine robot design. Specifically, the major difficulty of using CEM for optimizing robots in EvoGym is that the design space of EvoGym is discrete. We choose to let CEM optimize the probabilities of robot cell types ($P\in \mathbb{R}^{25\times 5}$), and take the types with maximum probabilities (`P.argmax(dim=1)`).
> >
> > For a fair comparison, the control policies of all the variants in ablation studies adopt the same network architecture and optimization algorithm (PPO) as our method HERD.
> >
> > We have provided the pseudo-code of the ablation variants in Algorithm 3 and 4 in Appendix B.4. Please refer to the revised version of our paper for details.
> >
> >
> > > **Question 3**: How is the performance of a HandCrafted robot determined？ Does it use a transformer-PPO policy? If yes, it should also be reported with variance, e.g., in Figure 3.
> >
> > **A**: The performance of the HandCrafted Robot is obtained by averaging the final performance of four runs with different random seeds. It also uses a transformer-PPO policy. The reason why we did not show its learning curve is that it is not a robot design method. HandCrafted Robot has a fixed human-designed robot morphology and only needs to learn control policy. It is unfair to directly compare this baseline with other robot design methods, whose robot morphologies are learned online. In this paper, we use the final performance improvement of our method over this baseline to show the necessity of creating robot design methods.
> >
> > Nevertheless, as suggested by the reviewer, for a complete comparison, we also include the learning curve as well as its variance of HandCrafted Robot in Figure 24 of Appendix C. Our method is much better than HandCrafted Robot in 7 tasks and has comparable performance in 5 tasks. Please refer to the revised version of our paper for this new result.
> >
> >
> > Thanks again for your efforts and valuable comments! We hope our explanations address your concerns and sincerely appreciate it if you could re-evaluate our work. Any further feedback and discussions are much appreciated.
> >
> > ---
> >
> > **Reference**
> >
> > [1] Moss, Robert J. "Cross-entropy method variants for optimization." arXiv preprint arXiv:2009.09043 (2020).

---

> > ### Comment · Reviewer_WvQM · 2023-11-22
> >
> > >A: The baselines CuCo, Vanilla RL, and HandCrafted Robot adopt the same control policy architecture (transformers) and learning method (PPO) as our method HERD. For the other baselines GA, BO, and CPPN-NEAT, we use their official implementations from CuCo and EvoGym.
> >
> > Thanks for the clarification!
> >
> > I think it is important to mention that the control policy of other baselines GA, BO, and CPPN-NEAT is **not the same** as other baselines and your method.

---

> > > ### Comment · Reviewer_WvQM · 2023-11-22
> > >
> > > All my concerns are addressed.
> > >
> > > I raised my score to Rating: 6.

---

> > > > ### Author Response · Authors · 2023-11-22
> > > > **Reply to Reviewer WvQM**
> > > >
> > > > Thank you for your timely reply and for raising the score. We also thank you for the constructive suggestion. We have added more details about the difference in control policy learning for baselines GA, BO, and CPPN-NEAT and other baselines and our method in Appendix B.3 in the revised version of our paper.

---

### Meta-Review · Area_Chair_JHhr · 2023-12-14

**Metareview:**

This paper proposes an algorithm for the automatic design of multi-cellular robots through the use of hyperbolic embeddings.

All the reviewers agree that this manuscript present a novel contribution and that the paper is well written. There are also no substantial concerns about the experimental evaluation.

One potential concern is the relevance for the wider ML community, given the narrow applicability of the proposed algorithm.

**Justification For Why Not Higher Score:**

Although all the reviewers agree that the paper have meaningful contributions. They all maintain some doubts after the rebuttal. Moreover, the contribution is very specific to a subfield and not of general interest to the larger ML community.

**Justification For Why Not Lower Score:**

All the reviewers agree that the paper have meaningful contributions.

---

### Decision · Program_Chairs · 2024-01-16

Accept (poster)